# SPARSE MIXTURE-OF-EXPERTS ARE DOMAIN GENERALIZABLE LEARNERS

**Bo Li**[1]* **Yifei Shen**[2]* **Jingkang Yang**[1] **Yezhen Wang**[3] **Jiawei Ren**[1]
**Tong Che**[3,4] **Jun Zhang**[2] **Ziwei Liu**[1]✉

[1]S-Lab, Nanyang Technological University
[2]The Hong Kong University of Science and Technology
[3]Mila-Quebec AI Institute [4]Nvidia Research
{libo0013,ziwei.liu}@ntu.edu.sg

## ABSTRACT

Human visual perception can easily generalize to out-of-distributed visual data, which is far beyond the capability of modern machine learning models. Domain generalization (DG) aims to close this gap, with existing DG methods mainly focusing on the loss function design. In this paper, we propose to explore an orthogonal direction, i.e., the design of the backbone architecture. It is motivated by an empirical finding that transformer-based models trained with empirical risk minimization (ERM) outperform CNN-based models employing state-of-the-art (SOTA) DG algorithms on multiple DG datasets. We develop a formal framework to characterize a network's robustness to distribution shifts by studying its architecture's alignment with the correlations in the dataset. This analysis guides us to propose a novel DG model built upon vision transformers, namely *Generalizable Mixture-of-Experts (GMoE)*. Extensive experiments on DomainBed demonstrate that GMoE trained with ERM outperforms SOTA DG baselines by a large margin. Moreover, GMoE is complementary to existing DG methods and its performance is substantially improved when trained with DG algorithms.

## 1 INTRODUCTION

### 1.1 MOTIVATIONS

Generalizing to out-of-distribution (OOD) data is an innate ability for human vision, but highly challenging for machine learning models (Recht et al., 2019; Geirhos et al., 2021; Ma et al., 2022). Domain generalization (DG) is one approach to address this problem, which encourages models to be resilient under various distribution shifts such as background, lighting, texture, shape, and geographic/demographic attributes.

From the perspective of representation learning, there are several paradigms towards this goal, including domain alignment (Ganin et al., 2016; Hoffman et al., 2018), invariant causality prediction (Arjovsky et al., 2019; Krueger et al., 2021), meta-learning (Bui et al., 2021; Zhang et al., 2021c), ensemble learning (Mancini et al., 2018; Cha et al., 2021b), and feature disentanglement (Wang et al., 2021; Zhang et al., 2021b). The most popular approach to implementing these ideas is to design a specific loss function. For example, DANN (Ganin et al., 2016) aligns domain distributions by adversarial losses. Invariant causal prediction can be enforced by a penalty of gradient norm (Arjovsky et al., 2019) or variance of training risks (Krueger et al., 2021). Meta-learning and domain-specific loss functions (Bui et al., 2021; Zhang et al., 2021c) have also been employed to enhance the performance. Recent studies have shown that these approaches improve ERM and achieve promising results on large-scale DG datasets (Wiles et al., 2021).
Meanwhile, in various computer vision tasks, the innovations in backbone architectures play a pivotal role in performance boost and have attracted much attention (He et al., 2016; Hu et al., 2018; Liu et al., 2021). Additionally, it has been empirically demonstrated in Sivaprasad et al. (2021) that different CNN architectures have different performances on DG datasets. Inspired by these pioneering works, we conjecture that *backbone architecture design would be promising for DG*. To verify this intuition, we evaluate a transformer-based model and compare it with CNN-based architectures of equivalent

---

*Equal contribution. ✉Corresponding author.

computational overhead, as shown in Fig. 1(a). To our surprise, a vanilla ViT-S/16 (Dosovitskiy et al., 2021) trained with empirical risk minimization (ERM) outperforms ResNet-50 trained with SOTA DG algorithms (Cha et al., 2021b; Rame et al., 2021; Shi et al., 2021) on DomainNet, OfficeHome and VLCS datasets, despite the fact that both architectures have a similar number of parameters and enjoy close performance on in-distribution domains. We theoretically validate this effect based on the algorithmic alignment framework (Xu et al., 2020a; Li et al., 2021). We first prove that a network trained with the ERM loss function is more robust to distribution shifts if its architecture is more *similar* to the invariant correlation, where the similarity is formally measured by the *alignment value* defined in Xu et al. (2020a). On the contrary, a network is less robust if its architecture aligns with the spurious correlation. We then investigate the alignment between backbone architectures (i.e., convolutions and attentions) and the correlations in these datasets, which explains the superior performance of ViT-based methods.

To further improve the performance, our analysis indicates that we should exploit properties of invariant correlations in vision tasks and design network architectures to align with these properties. This requires an investigation that sits at the intersection of domain generalization and classic computer vision. In domain generalization, it is widely believed that the data are composed of some sets of attributes and distribution shifts of data are distribution shifts of these attributes (Wiles et al., 2021). The latent factorization model of these attributes is almost identical to the generative model of visual attributes in classic computer vision (Ferrari & Zisserman, 2007). To capture these diverse attributes, we propose a Generalizable Mixture-of-Experts (GMoE), which is built upon sparse mixture-of-experts (sparse MoEs) (Shazeer et al., 2017) and vision transformer (Dosovitskiy et al., 2021). The sparse MoEs were originally proposed as key enablers for extremely large, but efficient models (Fedus et al., 2022). By theoretical and empirical evidence, we demonstrate that MoEs are experts for processing visual attributes, leading to a better alignment with invariant correlations. Based on our analysis, we modify the architecture of sparse MoEs to enhance their performance in DG. Extensive experiments demonstrate that GMoE achieves superior domain generalization performance both with and without DG algorithms.

## 1.2 CONTRIBUTIONS

In this paper, we formally investigate the impact of the backbone architecture on DG and propose to develop effective DG methods by backbone architecture design. Specifically, our main contributions are summarized as follows:

**A Novel View of DG:** In contrast to previous works, this paper initiates a formal exploration of the backbone architecture in DG. Based on algorithmic alignment (Xu et al., 2020a), we prove that a network is more robust to distribution shifts if its architecture aligns with the invariant correlation, whereas less robust if its architecture aligns with spurious correlation. The theorems are verified on synthetic and real datasets.

**A Novel Model for DG:** Based on our theoretical analysis, we propose Generalizable Mixture-of-Experts (GMoE) and prove that it enjoys a better alignment than vision transformers. GMoE is built upon sparse mixture-of-experts (Shazeer et al., 2017) and vision transformer (Dosovitskiy et al., 2021), with a theory-guided performance enhancement for DG.

**Excellent Performance:** We validate GMoE's performance on all 8 large-scale datasets of DomainBed. Remarkably, GMoE trained with ERM achieves SOTA performance on 7 datasets in the train-validation setting and on 8 datasets in the leave-one-domain-out setting. Furthermore, the GMoE trained with DG algorithms achieves better performance than GMoE trained with ERM.

## 2 PRELIMINARIES

### 2.1 NOTATIONS

Throughout this paper, $a$, $\boldsymbol{a}$, $\boldsymbol{A}$ stand for a scalar, a column vector, a matrix, respectively. $O(\cdot)$ and $\omega(\cdot)$ are asymptotic notations. We denote the training dataset, training distribution, test dataset, and test distribution as $\mathcal{E}_{tr}$, $D_{tr}$, $\mathcal{E}_{te}$, and $D_{te}$, respectively.

### 2.2 ATTRIBUTE FACTORIZATION

The attribute factorization (Wiles et al., 2021) is a realistic generative model under distribution shifts. Consider a joint distribution of the input $\boldsymbol{x}$ and corresponding attributes $a^1, \cdots, a^K$ (denoted as

$a^{1:K}$) with $a^i \in \mathcal{A}^i$, where $\mathcal{A}^i$ is a finite set. The label can depend on one or multiple attributes. Denote the latent factor as $z$, the data generation process is given by

$$z \sim p(z), \quad a^i \sim p(a^i|z), \quad x \sim p(x|z), \quad p(a^{1:K}, x) = p(a^{1:K}) \int p(x|z)p(z|a^{1:K})dz. \quad (1)$$

The distribution shift arises if different marginal distributions of the attributes are given but they share the same conditional generative process. Specifically, we have $p_{\text{train}}(a^{1:K}) \neq p_{\text{test}}(a^{1:K})$, but the generative model in equation 1 is shared across the distributions, i.e., we have $p_{\text{test}}(a^{1:K}, x) = p_{\text{test}}(a^{1:K}) \int p(x|z)p(z|a^{1:K})dz$ and similarly for $p_{\text{train}}$. The above description is abstract and we will illustrate with an example.

**Example 1.** (DSPRITES (Matthey et al., 2017)) Consider $\mathcal{A}^1 = \{\text{red, blue}\}$ and $\mathcal{A}^2 = \{\text{ellipse, square}\}$. The target task is a shape classification task, where the label depends on attribute $a^2$. In the training dataset, $90\%$ ellipses are red and $50\%$ squares are blue, while in the test dataset all the attributes are distributed uniformly. As the majority of ellipses are red, the classifier will use color as a shortcut in the training dataset, which is so-called *geometric skews* (Nagarajan et al., 2020). However, this shortcut does not exist in the test dataset and the network fails to generalize.

In classic computer vision, the attributes are named *visual attributes* and they follow a similar data generation process (Ferrari & Zisserman, 2007). We shall discuss them in detail in Section 4.2.

## 2.3 Algorithmic Alignment

We first introduce algorithmic alignment, which characterizes the easiness of IID reasoning tasks by measuring the similarity between the backbone architecture and target function. The alignment is formally defined as the following.

**Definition 1.** *(Alignment; (Xu et al., 2020a)) Let $\mathcal{N}$ denote a neural network with $n$ modules $\{\mathcal{N}_i\}_{i=1}^n$ and assume that a target function for learning $y = g(x)$ can be decomposed into $n$ functions $f_1, \cdots, f_n$. The network $\mathcal{N}$ aligns with the target function if replacing $\mathcal{N}_i$ with $f_i$, it outputs the same value as algorithm $g$. The alignment value between $\mathcal{N}$ and $f$ is defined as*

$$Alignment(\mathcal{N}, f, \epsilon, \delta) := n \cdot \max_i \mathcal{M}(f_i, \mathcal{N}_i, \epsilon, \delta), \quad (2)$$

*where $\mathcal{M}(f_i, \mathcal{N}_i, \epsilon, \delta)$ denotes the sample complexity measure for $\mathcal{N}_i$ to learn $f_i$ with $\epsilon$ precision at failure probability $\delta$ under a learning algorithm when the training distribution is the same as the test distribution.*

In Definition 1, the original task is to learn $f$, which is a challenging problem. Intuitively, if we could find a backbone architecture that is suitable for this task, it helps to break the original task into simpler sub-tasks, i.e., to learn $f_1, \cdots, f_n$ instead. Under the assumptions of algorithmic alignment (Xu et al., 2020a), $f$ can be learned optimally if the sub-task $f_1, \cdots, f_n$ can be learned optimally. Thus, a good alignment makes the target task easier to learn, and thus improves IID generalization, which is given in Theorem 3 in Appendix B.1. In Section 3, we extend this framework to the DG setting.

# 3 On the Importance of Neural Architecture for Domain Generalization

In this section, we investigate the impact of the backbone architecture on DG, from a motivating example to a formal framework.

## 3.1 A Motivating Example: CNNs versus Vision Transformers

We adopt DomainBed (Gulrajani & Lopez-Paz, 2021) as the benchmark, which implements SOTA DG algorithms with ResNet50 as the backbone. We test the performance of ViT trained with ERM on this benchmark, without applying any DG method. The results are shown in Fig. 1(a). To our surprise, ViT trained with ERM already outperforms CNNs with SOTA DG algorithms on several datasets, which indicates that the selection of the backbone architecture is potentially more important than the loss function in DG. In the remaining of this article, we will obtain a theoretical understanding of this phenomenon and improve ViT for DG by modifying its architecture.

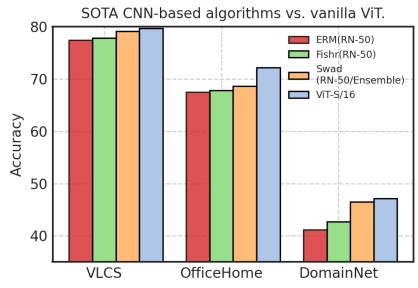

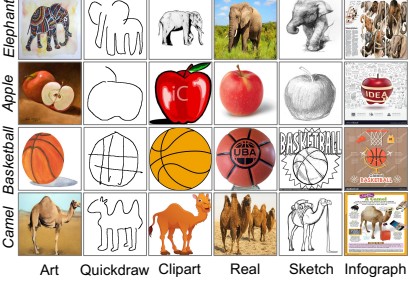

(a) Performance comparison.

(b) Dataset visualization.

Figure 1: (a) Performance comparison of ViT-S/16 (w/21.8M trainable parameters) with ERM and the ResNet-50 (w/25.6M) backbone with SOTA DG algorithms. ERM(RN-50), Fishr(RN-50), Swad(RN-50/Ensemble) denotes the ResNet-50 trained with ERM, Fish (Shi et al., 2021) , Fishr (Rame et al., 2021), and Swad (Cha et al., 2021b), respectively. (b) Image examples from DomainNet. Each row shows one class and each column shows one domain.

## 3.2 UNDERSTANDING FROM A THEORETICAL PERSPECTIVE

The above experiment leads to an intriguing question: how does the backbone architecture impact the network's performance in DG? In this subsection, we endeavor to answer this question by extending the algorithmic alignment framework (Xu et al., 2020a) to the DG setting.

To have a tractable analysis for nonlinear function approximation, we first make an assumption on the distribution shift.

**Assumption 1.** *Denote $\mathcal{N}_1$ as the first module of the network (including one or multiple layers) of the network. Let $p_{train,\mathcal{N}_1}(\boldsymbol{s})$ and $p_{test,\mathcal{N}_1}(\boldsymbol{s})$ denote the probability density functions of features after $\mathcal{N}_1$. Assume that the support of the training feature distribution covers that of the test feature distribution, i.e., $\max_{\boldsymbol{s}} \frac{p_{test,\mathcal{N}_1}(\boldsymbol{s})}{p_{train,\mathcal{N}_1}(\boldsymbol{s})} \leq C$, where $C$ is a constant independent of the number of training samples.*

**Remark 1.** (Interpretations of Assumption 1) This condition is practical in DG, especially when we have a pretrained model for disentanglement (e.g., on DomainBed (Gulrajani & Lopez-Paz, 2021)). In Example 1, the training distribution and test distribution have the same support. In DomainNet, although the elephants in quickdraw are visually different from the elephants in other domains, the quickdraw picture's attributes/features (e.g., big ears and long noise) are covered in the training domains. From a technical perspective, it is impossible for networks trained with gradient descent to approximate a wide range of nonlinear functions in the out-of-support regime (Xu et al., 2020b). Thus, this condition is necessary if we do not impose strong constraints on the target functions.

We define several key concepts in DG. The target function is an invariant correlation across the training and test datasets. For simplicity, we assume that the labels are noise-free.

**Assumption 2.** *(Invariant correlation) Assume there exists a function $g_c$ such that for training data, we have $g_c(\mathcal{N}_1(\boldsymbol{x})) = y, \forall x \in \mathcal{E}_{tr}$, and for test data, we have $\mathbb{P}_{D_{te}}[\|g_c(\mathcal{N}_1(\boldsymbol{x})) - y\| \leq \epsilon] > 1 - \delta$.*

We then introduce the spurious correlation (Wiles et al., 2021), i.e., some attributes are correlated with labels in the training dataset but not in the test dataset. The spurious correlation exists only if the training distribution differs from the test distribution and this distinguishes DG from classic PAC-learning settings (Xu et al., 2020a).

**Assumption 3.** *(Spurious correlation) Assume there exists a function $g_s$ such that for training data $g_s(\mathcal{N}_1(\boldsymbol{x})) = y, \forall x \in \mathcal{E}_{tr}$, and for test data, we have $\mathbb{P}_{D_{te}}[\|g_s(\mathcal{N}_1(\boldsymbol{x})) - y\| > \omega(\epsilon)] > 1 - \delta$.*

The next theorem extends algorithmic alignment from IID generalization (Theorem 3) to DG.

**Theorem 1.** *(Impact of Backbone Architecture in Domain Generalization) Denote $\mathcal{N}' = \{\mathcal{N}_2, \cdots, \mathcal{N}_n\}$. Assuming we train the neural network with ERM, and Assumption 1, 2, 3 hold, we have the following statements:*

    *1. If Alignment$(\mathcal{N}', g_c, \epsilon, \delta) \leq |\mathcal{E}_{tr}|$, we have $\mathbb{P}_{D_{te}}[\|\mathcal{N}(\boldsymbol{x}) - y\| \leq O(\epsilon)] > 1 - O(\delta)$;*

2. *If Alignment$(\mathcal{N}', g_s, \epsilon, \delta) \leq |\mathcal{E}_{tr}|$, we have $\mathbb{P}_{D_{te}}[\|\mathcal{N}(\boldsymbol{x}) - y\| > \omega(\epsilon)] > 1 - O(\delta)$.*

**Remark 2.** (Interpretations of Theorem 1) By choosing a sufficiently small $\epsilon$, only one of Alignment$(\mathcal{N}', g_c, \epsilon, \delta) \leq |\mathcal{E}_{tr}|$ and Alignment$(\mathcal{N}', g_s, \epsilon, \delta) \leq |\mathcal{E}_{tr}|$ holds. Thus, Theorem 1 shows that the networks aligned with invariant correlations are more robust to distribution shifts. In Appendix B.2, we build a synthetic dataset that satisfies all the assumptions. The experimental results exactly match Theorem 1. In practical datasets, the labels may have colored noise, which depends on the spurious correlation. Under such circumstances, the network should rely on multiple correlations to fit the label well and the correlation that best aligns with the network will have the major impact on its performance. Please refer to Appendix B.1 for the proof.

**ViT with ERM versus CNNs with DG algorithms**  We now use Theorem 1 to explain the experiments in the last subsection. The first condition of Theorem 1 shows that if the neural architecture aligns with the invariant correlation, ERM is sufficient to achieve a good performance. In some domains of OfficeHome or DomainNet, the shape attribute has an invariant correlation with the label, illustrated in Fig. 1(b). On the contrary, a spurious correlation exists between the attribute texture and the label. According to the analysis in Park & Kim (2022), multi-head attentions (MHA) are low-pass filters with a *shape* bias while convolutions are high-pass filters with a *texture* bias. As a result, a ViT simply trained with ERM can outperform CNNs trained with SOTA DG algorithms.

To improve ViT's performance, Theorem 1 suggests that we should exploit the properties of invariant correlations. In image recognition, objects are described by functional parts (e.g., visual attributes), with words associated with them (Zhou et al., 2014). The configuration of the objects has a large degree of freedom, resulting in different shapes among one category. Therefore, functional parts are more fundamental than shape in image recognition and we will develop backbone architectures to capture them in the next section.

## 4    GENERALIZABLE MIXTURE-OF-EXPERTS FOR DOMAIN GENERALIZATION

In this section, we propose Generalizable Mixture-of-Experts (GMoE) for domain generalization, supported by effective neural architecture design and theoretical analysis.

### 4.1    MIXTURE-OF-EXPERTS LAYER

In this subsection, we introduce the mixture-of-experts (MoE) layer, which is an essential component of GMoE. One ViT layer is composed of an MHA and an FFN. In the MoE layer, the FFN is replaced by mixture-of-experts and each expert is implemented by an FFN (Shazeer et al., 2017). Denoting the output of the MHA as $\boldsymbol{x}$, the output of the MoE layer with $N$ experts is given by

$$f_{\text{MoE}}(\boldsymbol{x}) = \sum_{i=1}^{N} G(\boldsymbol{x})_i \cdot E_i(\boldsymbol{x}) = \sum_{i=1}^{N} \text{TOP}_k(\text{Softmax}(\boldsymbol{W}\boldsymbol{x})) \cdot \boldsymbol{W}_{\text{FFN}_i}^2 \phi(\boldsymbol{W}_{\text{FFN}_i}^1 \boldsymbol{x}), \qquad (3)$$

where $\boldsymbol{W}$ is the learnable parameter for the gate, $\boldsymbol{W}_{\text{FFN}_i}^1$ and $\boldsymbol{W}_{\text{FFN}_i}^2$ are learnable parameters for the $i$-th expert, $\phi(\cdot)$ is a nonlinear activation function, and $\text{TOP}_k(\cdot)$ operation is a one-hot embedding that sets all other elements in the output vector as zero except for the elements with the largest $k$ values where $k$ is a hyperparameter. Given $\boldsymbol{x}_{\text{in}}$ as the input of the MoE layer, the update is given by

$$\boldsymbol{x} = f_{\text{MHA}}(\text{LN}(\boldsymbol{x}_{\text{in}})) + \boldsymbol{x}_{\text{in}}, \quad \boldsymbol{x}_{\text{out}} = f_{\text{MoE}}(\text{LN}(\boldsymbol{x})) + \boldsymbol{x},$$

where $f_{\text{MHA}}$ is the MHA layer, LN represents layer normalization, and $\boldsymbol{x}_{\text{out}}$ is the output of the MoE layer.

### 4.2    VISUAL ATTRIBUTES, CONDITIONAL STATEMENTS, AND SPARSE MOES

In real world image data, the label depends on multiple attributes. Capturing *diverse* visual attributes is especially important for DG. For example, the definition of an *elephant* in the Oxford dictionary is "a very large animal with thick grey skin, large ears, two curved outer teeth called tusks, and a long nose called a trunk". The definition involves three shape attributes (i.e., large ears, curved outer teeth, and a long nose) and one texture attribute (i.e., thick grey skin). In the IID ImageNet task, using the most discriminative attribute, i.e., the thick grey skin, is sufficient to achieve high accuracy (Geirhos

et al., 2018). However, in DomainNet, elephants no longer have grey skins while the long nose and big ears are preserved and the network relying on grey skins will fail to generalize.

The conditional statement (i.e., IF/ELSE in programming), as shown in Algorithm 1, is a powerful tool to efficiently capture the visual attributes and combine them for DG. Suppose we train the network to recognize the elephants on DomainNet, as illustrated in the first row of Fig. 1(b). For the elephants in different domains, shape and texture vary significantly while the visual attributes (large ears, curved teeth, long nose) are invariant across all the domains. Equipped with conditional statements, the recognition of the elephants can be expressed as "if an animal has large ears, two curved outer teeth, and a long nose, then it is an elephant". Then the subtasks are to recognize these visual attributes, which also requires conditional statements. For example, the operation for "curved outer teeth" is

---
**Algorithm 1:** `Conditional Statements`

---
Define intervals
$\quad I_i \subset \mathbb{R}, i = 1, \cdots, M$
Define functions
$\quad h_i, , i = 1, \cdots, M+1$
**switch** $h_1(\boldsymbol{x})$ **do**
$\quad$ **if** $h_1(\boldsymbol{x}) \in I_i$ **then**
$\quad\quad$ └ apply $h_{i+1}$ to $\boldsymbol{x}$

---

that "if the patch belongs to the teeth, then we apply a shape filter to it". In literature, the MoE layer is considered an effective approach to implement conditional computations (Shazeer et al., 2017; Riquelme et al., 2021). We formalize this intuition in the next theorem.

**Theorem 2.** *An MoE module in equation 3 with $N$ experts and $k = 1$ aligns with the conditional statements in Algorithm 1 with*

$$Alignment = \begin{cases} (N+1) \cdot \max\left(\mathcal{M}^*_{\mathscr{P}}, \mathcal{M}(G, h_1, \epsilon, \delta)\right), & \text{if } N < M, \\ (N+1) \cdot \max\left(\max\limits_{i \in \{1, \cdots, M\}} \mathcal{M}(f_{FFN_i}, h_{i+1}, \epsilon, \delta), \mathcal{M}(G, h_1, \epsilon, \delta)\right), & \text{if } N \geq M, \end{cases} \tag{4}$$

*where $\mathcal{M}(\cdot, \cdot, \cdot, \cdot)$ is defined in Definition 1, and $\mathcal{M}^*_{\mathscr{P}}$ is the optimal objective value of the following optimization problem:*

$$\begin{aligned} \mathscr{P} : &\underset{\mathcal{I}_1, \cdots \mathcal{I}_N}{minimize} \quad \underset{i \in \{1, \cdots, N\}}{\max} \quad \mathcal{M}(f_{FFN_i}, ([1_{I_j}]_{j \in \mathcal{I}_i} \circ h_1)^T \cdot [h_j]_{j \in \mathcal{I}_i}, \epsilon, \delta) \\ &subject\ to \quad \cup_{i=1}^N \mathcal{I}_i = \{2, 3, \cdots, M+1\}, \end{aligned} \tag{5}$$

*where $1_{I_j}$ is the indicator function on interval $I_j$.*

**Remark 3.** (Interpretations of Theorem 2) In algorithmic alignment, the network better aligns with the algorithm if the alignment value in equation 2 is lower. The alignment value between MoE and conditional statements depends on the product of $N + 1$ and a sample complexity term. When we increase the number of experts $N$, the alignment value first decreases as multiple experts decompose the original conditional statements into several simpler tasks. As we further increase $N$, the alignment value increases because of the factor $N + 1$ in the product. Therefore, the MoE aligns better with conditional statements than with the original FFN (i.e., $N = 1$). In addition, to minimize equation 5, similar conditional statements should be grouped together. By experiments in Section 5.4 and Appendix E.1, we find that sparse MoE layers are indeed experts for visual attributes, and similar visual attributes are handled by one expert. Please refer to Appendix B.3 for the proof.

### 4.3 ADAPTING MoE TO DOMAIN GENERALIZATION

In literature, there are several variants of MoE architectures, e.g., Riquelme et al. (2021); Fedus et al. (2022), and we should identify one for DG. By algorithmic alignment, in order to achieve a better generalization, the architecture of sparse MoEs should be designed to effectively handle visual attributes. In the following, we discuss our architecture design for this purpose.

**Routing scheme** Linear routers (i.e., equation 3) are often adopted in MoEs for vision tasks (Riquelme et al., 2021) while recent studies in NLP show that the cosine router achieves better performance in cross-lingual language tasks (Chi et al., 2022). For the cosine router, given input $\boldsymbol{x} \in \mathbb{R}^d$, the embedding $\boldsymbol{W}\boldsymbol{x} \in \mathbb{R}^{d_e}$ is first projected onto a hypersphere, followed by multiplying a learned embedding $\boldsymbol{E} \in \mathbb{R}^{d_e \times N}$. Specifically, the expression for the gate is given by

$$G(\boldsymbol{x}) = \text{TOP}_k\left(\text{Softmax}\left(\frac{\boldsymbol{E}^T \boldsymbol{W}\boldsymbol{x}}{\tau \|\boldsymbol{W}\boldsymbol{x}\| \|\boldsymbol{E}\|}\right)\right),$$

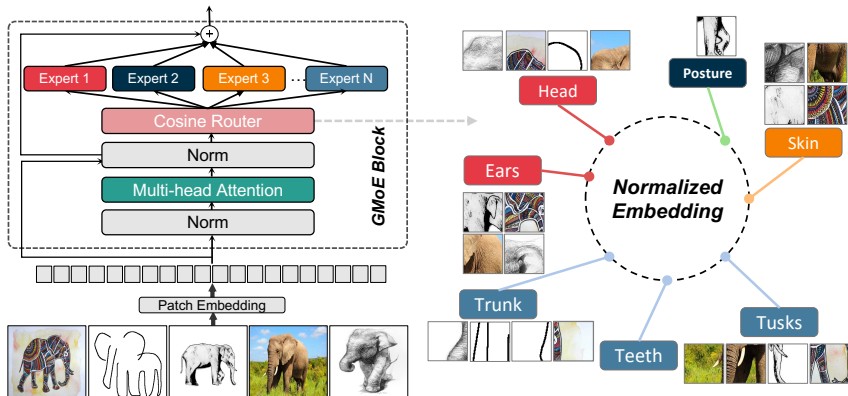

Figure 2: Overview architecture of GMoE. The cosine router distributes normalized image patches of different visual attributes to corresponding experts. Our analysis and experiments (in Section 4.2 and Section 5.4) demonstrate that an expert is potentially responsible for a group of similar visual attributes.

where $\tau$ is a hyper-parameter. In the view of image processing, $\boldsymbol{E}$ can be interpreted as the cookbook for visual attributes (Ferrari & Zisserman, 2007; Zhou et al., 2014) and the dot product between $\boldsymbol{E}$ and $\boldsymbol{W}\boldsymbol{x}$ with $\ell_2$ normalization is a matched filter. We opine that the linear router would face difficulty in DG. For example, the elephant image (and its all patches) in the Clipart domain is likely more similar to other images in the Clipart domain than in other domains. The issue can be alleviated with a codebook for visual attributes and matched filters for detecting them. Please refer to Appendix D.6 for the ablation study.

**Number of MoE layers** *Every-two* and *last-two* are two commonly adopted placement methods in existing MoE studies (Riquelme et al., 2021; Lepikhin et al., 2021). Specifically, every-two refers to replacing the even layer's FFN with MoE, and last-two refers to placing MoE at the last two even layers. For IID generalization, every-two often outperforms last-two (Riquelme et al., 2021). We argue that last-two is more suitable for DG as the conditional sentences for processing visual attributes are high-level. From experiments, we empirically find that last-two achieves better performance than every-two with fewer computations. Please refer to Appendix C.1 for more discussions and Appendix D.6 for the ablation study.

The overall backbone architecture of GMoE is shown in Fig. 2. To train diverse experts, we adopt the perturbation trick and load balance loss as in Riquelme et al. (2021). Due to space limitation, we leave them in Appendix C.4.

## 5 EXPERIMENTAL RESULTS

In this section, we evaluate the performance of GMoE on large-scale DG datasets and present model analysis to understand GMoE.

### 5.1 DOMAINBED RESULTS

In this subsection, we evaluate GMoE on DomainBed (Gulrajani & Lopez-Paz, 2021) with 8 benchmark datasets: PACS, VLCS, OfficeHome, TerraIncognita, DomainNet, SVIRO, Wilds-Camelyon and Wilds-FMOW. Detailed information on datasets and evaluation protocols are provided in Appendix D.1. The experiments are averaged over 3 runs as suggested in (Gulrajani & Lopez-Paz, 2021).

We present results in Table 1 with **train-validation selection**, which include baseline methods and recent DG algorithms and GMoE trained with ERM. The results demonstrate that GMoE without DG algorithms already outperforms counterparts on almost all the datasets. Meanwhile, GMoE has excellent performance in **leave-one-domain-out criterion**, and we leave the results in Appendix D.3 due to space limit. In the lower part of Table 1, we test our methods on three large-scale datasets: SVIRO, Wilds-Camelyon, and Wilds-FMOW. The three datasets capture real-world distribution shifts

Table 1: Overall out-of-domain accuracies with train-validation selection criterion. The best result is highlighted in **bold**. GMoE achieves the **best** performance on PACS, VLCS, OfficeHome, and DomainNet while ranking in the third-best on TerraIncognita.

| Algorithm | PACS | VLCS | OfficeHome | TerraInc | DomainNet |
|---|---|---|---|---|---|
| ERM (ResNet50) (Vapnik, 1991) | $85.7 \pm 0.5$ | $77.4 \pm 0.3$ | $67.5 \pm 0.5$ | $47.2 \pm 0.4$ | $41.2 \pm 0.2$ |
| IRM [ArXiv 20] (Arjovsky et al., 2019) | $83.5 \pm 0.8$ | $78.5 \pm 0.5$ | $64.3 \pm 2.2$ | $47.6 \pm 0.8$ | $33.9 \pm 2.8$ |
| DANN [JMLR 16] (Ganin et al., 2016) | $84.6 \pm 1.1$ | $78.7 \pm 0.3$ | $68.6 \pm 0.4$ | $46.4 \pm 0.8$ | $41.8 \pm 0.2$ |
| CORAL [ECCV 16] (Sun & Saenko, 2016) | $86.0 \pm 0.2$ | $77.7 \pm 0.5$ | $68.6 \pm 0.4$ | $46.4 \pm 0.8$ | $41.8 \pm 0.2$ |
| MMD [CVPR 18] (Li et al., 2018b) | $85.0 \pm 0.2$ | $76.7 \pm 0.9$ | $67.7 \pm 0.1$ | $42.2 \pm 1.4$ | $39.4 \pm 0.8$ |
| FISH [ICLR 22] (Shi et al., 2021) | $85.5 \pm 0.3$ | $77.8 \pm 0.3$ | $68.6 \pm 0.4$ | $45.1 \pm 1.3$ | $42.7 \pm 0.2$ |
| SWAD [NeurIPS 21] (Cha et al., 2021a) | $88.1 \pm 0.1$ | $79.1 \pm 0.1$ | $70.6 \pm 0.2$ | $50.0 \pm 0.3$ | $46.5 \pm 0.1$ |
| Fishr [ICML 22] (Rame et al., 2021) | $85.5 \pm 0.2$ | $77.8 \pm 0.2$ | $68.6 \pm 0.2$ | $47.4 \pm 1.6$ | $41.7 \pm 0.0$ |
| MIRO [ECCV 22] (Cha et al., 2022) | $85.4 \pm 0.4$ | $79.0 \pm 0.0$ | $70.5 \pm 0.4$ | $\mathbf{50.4 \pm 1.1}$ | $44.3 \pm 0.2$ |
| ERM (ViT-S/16) [ICLR 21] (Dosovitskiy et al., 2021) | $86.2 \pm 0.1$ | $79.7 \pm 0.0$ | $72.2 \pm 0.4$ | $42.0 \pm 0.8$ | $47.3 \pm 0.2$ |
| **GMoE-S/16 (Ours)** | $\mathbf{88.1 \pm 0.1}$ | $\mathbf{80.2 \pm 0.2}$ | $\mathbf{74.2 \pm 0.4}$ | $\underline{48.5} \pm 0.4$ | $\mathbf{48.7 \pm 0.2}$ |

| Algorithms | SVIRO | Wilds-Camelyon | Wilds-FMOW |
|---|---|---|---|
| ERM (ResNet50) (Vapnik, 1991) | $85.7 \pm 0.1$ | $93.1 \pm 0.2$ | $40.6 \pm 0.4$ |
| ERM (ViT-S/16) [ICLR 21] (Dosovitskiy et al., 2021) | $89.6 \pm 0.0$ | $91.1 \pm 0.1$ | $44.8 \pm 0.2$ |
| **GMoE-S/16 (Ours)** | $\mathbf{90.3 \pm 0.1}$ | $\mathbf{93.7 \pm 0.2}$ | $\mathbf{46.6 \pm 0.4}$ |

across a diverse range of domains. We adopt the data preprocessing and domain split in DomainBed. As there is no previous study conducting experiments on these datasets with DomainBed criterion, we only report the results of our methods, which reveal that GMoE outperforms the other two baselines.

## 5.2 GMoE with DG Algorithms

GMoE's generalization ability comes from its internal backbone architecture, which is orthogonal to existing DG algorithms. This implies that the DG algorithms can be applied to improve the GMoE's performance. To validate this idea, we apply two DG algorithms to GMoE, including one modifying loss functions approaches (Fish) and one adopting model ensemble (Swad). The results in Table 2 demonstrate that adopting GMoE instead of ResNet-50 brings significant accuracy promotion to these DG algorithms. Experiments on GMoE with more DG algorithms are in Appendix D.4.

## 5.3 Single-Source Domain Generalization Results

In this subsection, we create a challenging task, *single-source domain generalization*, to focus on generalization ability of backbone architecture. Specifically, we train the model only on data from one domain, and then test the model on multiple domains to validate its performance across all domains. This is a challenging task as we cannot rely on multiple domains to identify invariant correlations, and popular DG algorithms cannot be applied. We compare several models mentioned in above analysis (*e.g.,* ResNet, ViT, GMoE) with different scale of parameters, including their float-point-operations per

Table 2: GMoE trained with DG algorithms.

| Algorithm | DomainNet |
|---|---|
| GMoE | 48.7 |
| Fish (Rame et al., 2021) | 42.7 |
| **GMoE w/Fish** | **48.8** |
| Swad (Cha et al., 2021a) | 46.5 |
| **GMoE w/Swad** | **49.6** |

second (flops), IID and OOD accuracy. From the results in Table 3, we see that GMoE's OOD generalization gain over ResNet or ViT is much larger than that in the IID setting, which shows that GMoE is suitable for challenging domain generalization. Due to space limitation, we leave experiments with other training domains in Appendix D.5.

## 5.4 Model Analysis

In this subsection, we present diagnostic datasets to study the connections between MoE layer and the visual attributes.

**Diagnostic datasets: CUB-DG** We create CUB-DG from the original Caltech-UCSD Birds (CUB) dataset (Wah et al., 2011). We stylize the original images into another three domains, Candy, Mosaic and Udnie. The examples of CUB-DG are presented in Fig. 3(a). We evaluate GMoE and other DG algorithms that address domain-invariant representation (e.g., DANN). The results are in Appendix E.1, which demonstrate superior performance of GMoE compared with other DG algorithms. CUB-DG datasets provides rich visual attributes (e.g. beak's shape, belly's color) for each

Table 3: Single-source DG accuracy (%). Models are trained on Paint domain and tested (1) on Paint validation set (2) the rest 5 domains' validation sets on DomainNet. The flops are reference values to compare the model's computational efficiency. **IID Imp.** denotes IID improvement on Paint's validation set comparing with ResNet50. **OOD Imp.** denotes average OOD improvement across 5 test domains.

| MACs | Paint | Clipart | Info | Paint | Quick | Real | Sketch | IID Imp. | OOD Imp. |
|------|-------|---------|------|-------|-------|------|--------|----------|----------|
| 4.1G | ResNet50 | 37.1 | 12.9 | 62.7 | 2.2 | 49.3 | 33.3 | - | - |
| 7.9G | ResNet101 | 40.5 | 13.1 | 63.4 | 3.1 | 51.2 | 35.4 | 1.1% | 12.4% |
| 4.6G | ViT-S/16 | 42.7 | 15.9 | 69.0 | 5.0 | **56.4** | 37.0 | 10.0% | 38.6% |
| 4.8G | GMoE-S/16 | **43.5** | **16.1** | **69.3** | **5.3** | **56.4** | **38.0** | **10.5%** | **42.3%** |

image. That additional information enables us to measure the correlation between visual attributes and the router's expert selection.

**Visual Attributes & Experts Correlation**   We choose GMoE-S/16 with 6 experts in each MoE layer. After training the model on CUB-DG, we perform forward passing with training images and save the routers top-1 selection. Since the MoE model routes patches instead of images and CUB-DG has provided the location of visual attributes, we first match a visual attribute with its nearest 9 patches, and then correlate the visual attribute and its current 9 patches' experts. In Fig. 3(b), we show the 2D histogram correlation between the selected experts and attributes. Without any supervision signal for visual attributes, the ability to correlate attributes automatically emerges during training. Specifically, 1) each expert focus on distinct visual attributes; and 2) similar attributes are attended by the same expert (e.g., the left wing and the right wing are both attended by e4). This verifies the predictions by Theorem 2.

**Expert Selection**   To further understand the expert selections of the entire image, we record the router's selection for each patch (in GMoE-S/16, we process an image into $16 \times 16$ patches), and then visualize the top-1 selections for each patch in Fig. 3(c). In this image, we mark the selected expert for each patch with a black number and draw different visual attributes of the bird (*e.g.,* beak and tail types) with large circles. We see the consistent relationship between Fig. 3(b) and Fig. 3(c). In detail, we see (1) experts 0 and 2 are consistently routed with patches in the *background* area; (2) the *left wing*, *right wing*, *beak* and *tail* areas are attended by expert 3; (3) the *left leg* and *right leg* areas are attended by expert 4. More examples are given in Appendix E.1.

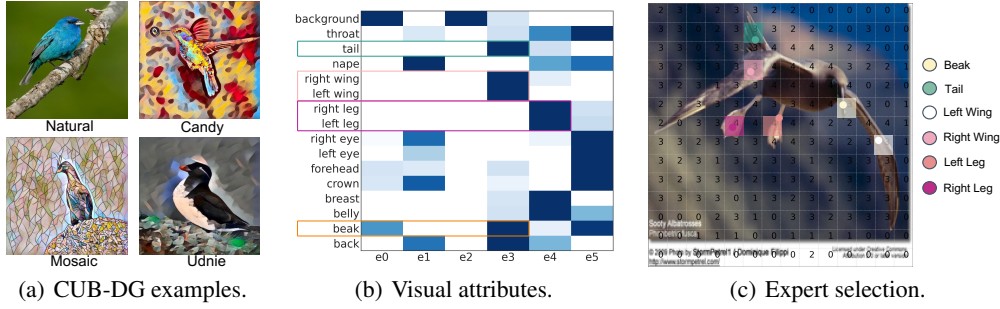

(a) CUB-DG examples.          (b) Visual attributes.          (c) Expert selection.

Figure 3: (a) Examples of CUB-DG datasets from four domains. (b)The $y$-axis corresponds to $15 + 1$ attributes (15 visual attributes + 1 background). The $x$-axis corresponds to the selected expert id. (c) Finetuned GMoE's router decision of block-10. The image is from CUB-DG's natural domain.

## 6   CONCLUSIONS

This paper is an initial step in exploring the impact of the backbone architecture in domain generalization. We proved that a network is more robust to distribution shifts if its architecture aligns well with the invariant correlation, which is verified on synthetic and real datasets. Based on our theoretical analysis, we proposed GMoE and demonstrated its superior performance on DomainBed. As for future directions, it is interesting to develop novel backbone architectures for DG based on algorithmic alignment and classic computer vision.

## 7 ETHICS STATEMENTS

Our paper suggests a new direction for Domain Generalization. We think this contribution can direct people to critically select model architecture in situations that require enhancing the model's out-of-domain generalization ability. We validated our conclusions under certain assumptions mentioned in the paper and empirically validated our findings in the image classification scenario. However, when applying our model or conclusions to other scenarios and specific usages, they still need careful validation before being actually applied. Besides, we did not use any non-public data, unauthorized software, or API in our paper, there are no privacy or other related ethical concerns.

## 8 ACKNOWLEDGMENTS

This research/project is supported by the National Research Foundation, Singapore under its AI Singapore Programme (AISG Award No: AISG2-PhD-2022-01-029). Besides, this project is also supported by NTU NAP, MOE AcRF Tier 2 (MOE-T2EP20221-0012), and under the RIE2020 Industry Alignment Fund – Industry Collaboration Projects (IAF-ICP) Funding Initiative, as well as cash and in-kind contribution from the industry partner(s).

We also thank Yushi Lan, Quanzhou Li, Ziqi Huang, Siyao Li, Runqi Pan, and Wenzhen Zhu for their help in revising the manuscript. Any opinions, findings, and conclusions or recommendations expressed in this material are those of the authors and do not necessarily reflect the views, policies, or endorsements, either expressed or implied, of NTU or the Singapore Government.

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

CONTENTS

## A RELATED WORKS

### A.1 DOMAIN GENERALIZATION

Domain Generalization (DG) aims to maintain the good performance of machine learning models even in the domains that are different from the training (source) domain. The following are the categories of mainstream domain generalization research.

**(1) Invariant Learning**: Aligning domain distributions and finding invariance across domains has been often studied with empirical results and theoretical proofs (Ganin et al., 2016; Zhao et al., 2019a). Specifically, researchers have explicitly sought aligning feature distributions based on the maximum mean discrepancy (MMD) (Li et al., 2018b), second order correlation (Sun & Saenko, 2016), moment matching (Peng et al., 2019), etc. Besides aligning feature distributions, Arjovsky et al. (Arjovsky et al., 2019) proposed IRM to learn an ideal invariant classifier on top of the representation space, which has inspired many follow-up works (Krueger et al., 2021; Ahuja et al., 2021; Zhou et al., 2022). These studies implement invariant learning via loss function designs. In this paper, we show that if we have a suitable backbone architecture, ERM is sufficient to find the invariant correlations, which indicates that the backbone architecture is as important as loss function design.

**(2) Ensemble Learning and Meta-learning**: Apart from learning invariant features and correlations across the domain, model-specific or domain-specific information helps improve the performance on DG datasets. The ensemble learning methods combine different models to exploit model-specific information (Mancini et al., 2018; Segù et al., 2020; Arpit et al., 2021; Li et al., 2022). In addition,

SWAD (Cha et al., 2021b) inhibits models from being overfit to local sharp minima by averaging model weights below a validation loss threshold. Meta-learning-based approaches (Li et al., 2018a; 2019; Dou et al., 2019; Zhang et al., 2021c; Bui et al., 2021) leverage domain-specific feature to improve the accuracy on each domain.

**(3) Data Manipulation**: Diverse training data are helpful for improving generalization, and researchers have proposed different manipulation/augmentation techniques (Nazari & Kovashka, 2020; Riemer et al., 2019), domain randomization (Yue et al., 2019; Zakharov et al., 2019). Furthermore, a line of works (Qiao et al., 2020; Liu et al., 2018; Zhao et al., 2019b) have exploited generating data samples to enhance the model generalization ability.

**(4) Neural Architecture Search**: Recently, neural architecture search-based methods were employed for DG. In Zhang et al. (2021a), the lottery ticket hypothesis was extended to DG settings: a biased full network contains an unbiased subnetwork that can achieve better OOD performance. The network pruning method was further adopted to identify such subnetworks. Differentiable architecture search was employed in Bai et al. (2021) to find a good CNN architecture for DG. These works are independent of backbone architecture design.

One unique advantage of the proposed method is that it is orthogonal to the above approaches, meaning that the performance of GMoE might be enhanced when combined with these approaches.

## A.2 VISION TRANSFORMERS

Originated from the machine translation tasks, Transformer (Vaswani et al., 2017) has recently received great attention in computer vision (Dosovitskiy et al., 2021; Liu et al., 2021; Wang et al., 2022) for their unprecedented performance in image recognition, semantic segmentation, and visual question answering. In addition to strong performance, some recent works empirically demonstrate the robustness of the ViTs over CNNs in terms of adversarial noise (Qin et al., 2021) and distribution shifts (Zhang et al., 2022). Nevertheless, the theoretical understandings remain elusive and this paper theoretically validates these effects. The analysis also motivates us to employ mixture-of-experts to enhance the performance of ViT models.

## A.3 SPARSE MIXTURE-OF-EXPERTS

Mixture-of-Experts models, or MoEs, make use of the outputs from several sub-models (experts) through an input-dependent routing mechanism for stronger model performance (Jacobs et al., 1991; Jordan & Jacobs, 1994). This training paradigm has led to the development of a plethora of methods for a wide ranges of applications (Hu et al., 1997; Tani & Nolfi, 1999). However, integrating MoEs and big models will inevitably introduce even larger model sizes and longer inference time. Sparse MoEs (Shazeer et al., 2017) were proposed with their routers to select only a few experts so that the inference time is on-par with the standalone counterpart. They were considered promising ways to scale up vision models (Riquelme et al., 2021). Some existing studies have applied MoE-type architecture to generalization tasks. (Guo et al., 2018) considers the muti-source domain adaptation in NLP, where one classifier is trained on each domain and a MoE is adopted to ensemble the classifiers. (Rahaman et al., 2021) investigates the systematic generalization problem and proposes to dynamically compose experts instead of selecting experts in MoEs. In this paper, we consider the domain generalization (DG) problem and prove that a backbone more aligned to invariant correlations is more robust to distribution shift. Based on the theory, we repurpose MoE for DG tasks and make several modifications for performance enhancement.

## A.4 OUT-OF-DISTRIBUTION GENERALIZATION THEORY

The classic PAC learning (Valiant, 1984) is only applicable to IID settings and recently there have been a line of works to investigate the theory of OOD generalization, including structured causal models (SCM) (Arjovsky et al., 2019; Ahuja et al., 2021; Zhou et al., 2022) and extrapolation theory (Xu et al., 2020b; Ziyin et al., 2020; Ye et al., 2021). The SCM-based approaches often assume that the invariant correlation is linear (Arjovsky et al., 2019; Ahuja et al., 2021; Zhou et al., 2022). The extrapolation theory focused on the situation where the supports of training distribution and test distribution are different (Xu et al., 2020b; Ziyin et al., 2020; Ye et al., 2021), where the networks have very limited ability for nonlinear function approximation (Xu et al., 2020b). Different from

existing works, this paper studies the situation where the invariant correlation is an arbitrary nonlinear function, and one of our main focuses is to justify why these assumptions hold in DG of vision tasks.

## A.5 ALGORITHMIC ALIGNMENT

The algorithmic alignment framework (Xu et al., 2020a) was originally proposed to understand the effectiveness of specialized network structures in reasoning tasks. Specifically, it characterizes the generalization bound of reasoning tasks by studying the alignment between the reasoning algorithm and the computation graph of the neural network. The underlying intuition is that if they align well, the neural network only needs to learn *simple* functions, which leads to better sample efficiency. This framework was further extended in Li et al. (2021) to investigate the impact of backbone architectures on the robustness of noisy labels. This paper adopts algorithmic alignment to provide a novel perspective of DG and develops a new architecture based on it.

## A.6 EMPIRICAL INVESTIGATIONS OF DISTRIBUTION SHIFT

Prior to this work, some empirical studies (Wiles et al., 2021; Sivaprasad et al., 2021) systematically investigate the impact of the data argumentation, the optimizer, the backbone architecture, and DG algorithms on domain generalization datasets, drawing various conclusions. Nevertheless, they did not explain when a specific model will succeed in DG and how to design the backbone architecture to improve DG performance. As far as we know, our paper is an initial attempt to theoretically investigate these questions, which provides a novel perspective of DG. In this paper, we prove that a backbone is more robust to distribution shift if it aligns with the invariant correlation in the dataset and a better alignment leads to better performance in DG. A novel architecture is designed based on our theory. We believe these results will encourage further research in this direction.

# B THEOREMS AND PROOFS

## B.1 PROOF OF THEOREM 1

We first state the algorithmic alignment theorem in the IID setting and then extend it to the DG setting.

**Theorem 3.** *(Alignment improves IID generalization; (Xu et al., 2020a)) Fix $\epsilon$ and $\delta$. Given a target function $g$ and a neural network $\mathcal{N}$, suppose $\{\boldsymbol{x}_i\}_{i=1}^M$ are i.i.d. samples drawn from some distribution $D$, and let $y_i = g(\boldsymbol{x}_i)$. Assumptions:*

*(a) Algorithm stability. Let $A$ be a learning algorithm for $\mathcal{N}_i$'s. Suppose $f = A(\{\boldsymbol{x}_i, y_i\})$ and $\hat{f} = A(\{\hat{\boldsymbol{x}}_i, y_i\})$. For any $\boldsymbol{x}$, $\|f(\boldsymbol{x}) - \hat{f}(\boldsymbol{x})\| \le L_0 \cdot \max_i \|x_i - \hat{x}_i\|$, where $x_i$ is the $i$-th coordinate of $\boldsymbol{x}$.*

*(b) Sequential learning. We train $\mathcal{N}_i$ sequentially: $\mathcal{N}_1$ has input samples $\{\boldsymbol{x}_i^{(1)}, f_1(\boldsymbol{x}_i^{(1)})\}_{i=1}^N$, with $\boldsymbol{x}_i^{(1)}$ obtained from the training dataset. For $j > 1$, the inputs $\hat{\boldsymbol{x}}_i^{(j)}$ for $\mathcal{N}_j$ are the outputs from the previous modules, but labels are generated by the correct functions $f_{j-1}, \cdots, f_1$ on $\hat{\boldsymbol{x}}_i^{(1)}$.*

*(c) Lipschitzness. The learned functions $\hat{f}_j$ satisfy $\|\hat{f}_j(\boldsymbol{x}) - \hat{f}_j(\hat{\boldsymbol{x}})\| \le L_1 \|\boldsymbol{x} - \hat{\boldsymbol{x}}\|$ for some $L_1$.*

*Under assumptions (a)(b)(c), Alignment$(\mathcal{N}, g, \epsilon, \delta) \le M$ implies there is a learning algorithm $A$ such that*

$$\mathbb{P}_{\boldsymbol{x} \sim D}[\|\mathcal{N}(\boldsymbol{x}) - g(\boldsymbol{x})\| \le O(\epsilon)] \ge 1 - O(\delta),$$

*where $\mathcal{N}$ is the network generated by $A$ on the training data $\{\boldsymbol{x}_i, y_i\}_{i=1}^M$.*

**Remark 4.** (Explanations of Assumptions in Theorem 3) The first and third assumptions are common assumptions in machine learning and are practical. The second assumption is impractical as we do not have auxiliary labels. However, the same pattern is observed for end-to-end learning in experiments (Xu et al., 2020a; Li et al., 2021).

We now prove Theorem 3.

**Theorem 4.** *(Impact of Backbone Architecture in DG) Let $\mathcal{N}' = \{\mathcal{N}_2, \cdots, \mathcal{N}_n\}$. Assuming we train the neural network with ERM, and Assumption 1, 2, 3, we have the following statements:*

1. *If Alignment$(\mathcal{N}', g_c, \epsilon, \delta) \leq |\mathcal{E}_{tr}|$, we have $\mathbb{P}_{D_{te}}[\|\mathcal{N}(\boldsymbol{x}) - y\| \leq O(\epsilon)] > 1 - O(\delta)$;*

2. *If Alignment$(\mathcal{N}', g_s, \epsilon, \delta) \leq |\mathcal{E}_{tr}|$, we have $\mathbb{P}_{D_{te}}[\|\mathcal{N}(\boldsymbol{x}) - y\| > \omega(\epsilon)] > 1 - O(\delta)$.*

*Proof.* The proof is mainly based on Theorem 3. We tackle the distribution shift in $\boldsymbol{x}$ with Lemma 1 and analyze the distribution shift in $y$ with Assumption 2, 3.

**First condition:** From Theorem 3,

$$\mathbb{P}_{\boldsymbol{x} \sim D_{tr}}[\|\mathcal{N}(\boldsymbol{x}) - g_c(\mathcal{N}_1(\boldsymbol{x}))\| > O(\epsilon)] \leq O(\delta).$$

By Lemma 1, we have

$$\mathbb{P}_{\boldsymbol{x} \sim D_{te}}[\|\mathcal{N}(\boldsymbol{x}) - g_c(\mathcal{N}_1(\boldsymbol{x}))\| \leq O(\epsilon)] \leq C\mathbb{P}_{\boldsymbol{x} \sim D_{tr}}[\|\mathcal{N}(\boldsymbol{x}) - g_c(\boldsymbol{x})\| < O(\epsilon)] \leq O(\delta). \quad (6)$$

Combing with the second condition of Assumption 2, we have

$$\begin{aligned} &\mathbb{P}_{D_{te}}[\|\mathcal{N}(\boldsymbol{x}) - g(\boldsymbol{x})\| \leq O(\epsilon)] \\ \geq &\mathbb{P}_{D_{te}}[\|g_c(\mathcal{N}_1(\boldsymbol{x})) - g(\boldsymbol{x})\| \leq O(\epsilon)] \cdot \mathbb{P}_{D_{te}}[\|\mathcal{N}(\boldsymbol{x}) - g_c(\mathcal{N}_1(\boldsymbol{x}))\| \leq O(\epsilon)] \\ \geq &(1 - \delta)(1 - O(\delta)) = 1 - O(\delta), \end{aligned}$$

where the first inequality follows $O(\epsilon) + O(\epsilon) = O(\epsilon)$, and the second inequality follows the second condition of Assumption 2 and equation 6.

**Second condition:** Following the proof of equation 6, we obtain

$$\mathbb{P}_{D_{te}}[\|\mathcal{N}(\boldsymbol{x}) - g_s(\mathcal{N}_1(\boldsymbol{x}))\| \leq O(\epsilon)] \leq 1 - O(\delta). \quad (7)$$

Combing with the second condition of Assumption 3, we have

$$\begin{aligned} &\mathbb{P}_{D_{te}}[\|\mathcal{N}(\boldsymbol{x}) - g(\boldsymbol{x})\| > \omega(\epsilon)] \\ \geq &\mathbb{P}_{D_{te}}[\|g_s(\mathcal{N}_1(\boldsymbol{x})) - g(\boldsymbol{x})\| > \omega(\epsilon)] \cdot \mathbb{P}_{D_{te}}[\|\mathcal{N}(\boldsymbol{x}) - g_s(\mathcal{N}_1(\boldsymbol{x}))\| \leq O(\epsilon)] \\ \geq &(1 - \delta)(1 - O(\delta)) = 1 - O(\delta), \end{aligned}$$

where the first inequality follows $\omega(\delta) - O(\delta) = O(\delta)$, and the second inequality follows the second condition of Assumption 3 and equation 7. This finishes the proof for Theorem 1. $\square$

## B.2 Synthetic Experiments for Theorem 1

In this subsection, we set up synthetic experiments to illustrate Theorem 1.

### B.2.1 Noiseless Version

We first set up experiments following the assumptions in Theorem 1.

**Training dataset generation:** Consider data pairs $(\boldsymbol{x}, y) \in (\mathbb{R}^K)^P \times \{1, \cdots, K\}$ in the training and validation datasets generated as follows.

- Generate label $y \in \{1, \cdots, K\}$ uniformly.
- Generate $K$ mutually orthogonal feature vectors $\{\boldsymbol{c}_k\}$ such that $\boldsymbol{c}_k \in \mathbb{R}^K$ and $\|\boldsymbol{c}_k\|_2 = 1$.
- Generate $x$ as a collection of $P$ patches: $\boldsymbol{x} = (\boldsymbol{x}^{(1)}, \cdots, \boldsymbol{x}^{(P)}) \in (\mathbb{R}^d)^P$
  - **Pixel-level feature.** For the first patch, the $y$-th pixel is set as 1 and other pixels drawn from $N(0, 1)$.
  - **Patch-level feature.** Uniformly select one and only one patch given by $\boldsymbol{c}_y$.
  - **Random noise.** The rest patches are Gaussian noise drawn independently from $N(0, \boldsymbol{I}_K)$.

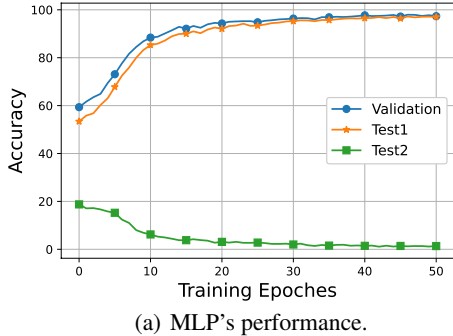
(a) MLP's performance.

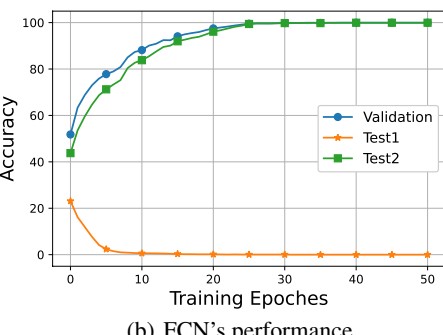
(b) FCN's performance.

Figure 4: Performance of the MLP and FCN on the synthetic dataset. MLP aligns with the invariant correlations in test dataset 1 while FCN aligns with the invariant correlations in test dataset 2.

The generation of the validation dataset follows that of the training dataset.

**Test dataset generation:** Two OOD test datasets are given. The first dataset differs from the training dataset in the pixel-level feature. Specifically, we set the $(y + 1) \mod K$-th pixel of the first patch as 1 and other pixels in the first patch follow $N(0, 1)$, i.e., making the pixel-level features spurious. The second dataset is distinguished from the training dataset such that the patch-level feature is $c_{(y+1) \mod K}$, i.e., making patch-level features spurious.

**Backbone architectures:** We adopt MLPs and fully convolutional networks (FCNs) as MLPs align with pixel-level features while FCNs align with patch-level features. It is clear that both MLPs and FCNs can fit the label only from pixel-level features or patch-level features.

**Parameters in experiments:** We set $P = 10$, $K = 4$. We generate $100,000$ samples for the training dataset and $2,000$ test samples for the validation and test datasets. We use a three-layer MLP with hidden size $\{40, 100, 100, 4\}$ and a two-layer FCN with 20 filters.

The experimental results are shown in Fig. 4. The MLP completely fails on the second dataset while the FCN completely fails on the first dataset, which is consistent with Theorem 1.

### B.2.2 NOISY VERSION

In the previous experiment, both pixel-level features and patch-level features can perfectly fit the label. We give a noisy version as follows:

1. **Pixel-level feature.** Given a probability value $p_1$. With probability $p_1$, the $y$-th pixel of the first patch is set as 1 and other pixels are drawn from $N(0, 1)$. With probability $1 - p_1$, the $y'$-th ($y' \neq y$) pixel of the first patch is set as 1 and other pixels are drawn from $N(0, 1)$.

2. **Patch-level feature.** Given a probability value $p_2$. With probability $p_2$, uniformly select one and only one patch given by $c_y$. With probability $1 - p_2$, uniformly select one and only one patch given by $c_{y'}$ ($y' \neq y$).

The experiments with different $p_1$ and $p_2$ are shown in Fig. 5. When the signal-to-noise ratios of pixel-level feature and patch-level feature are similar, the MLP learns the pixel-level feature, which corresponds to its alignment. It shows that the network prefers learning the correlation that aligns with its architecture, even when exploiting other correlations may lead to performance gains. When the difference between $p_1$ and $p_2$ is large, the network will still learn its aligned correlation at the early stage (with above $90\%$ accuracy on Test1), but may change when the number of epochs is large. This suggests that early stopping is beneficial to DG as suggested in Cha et al. (2021b) under the condition that the backbone aligns to some invariant correlations.

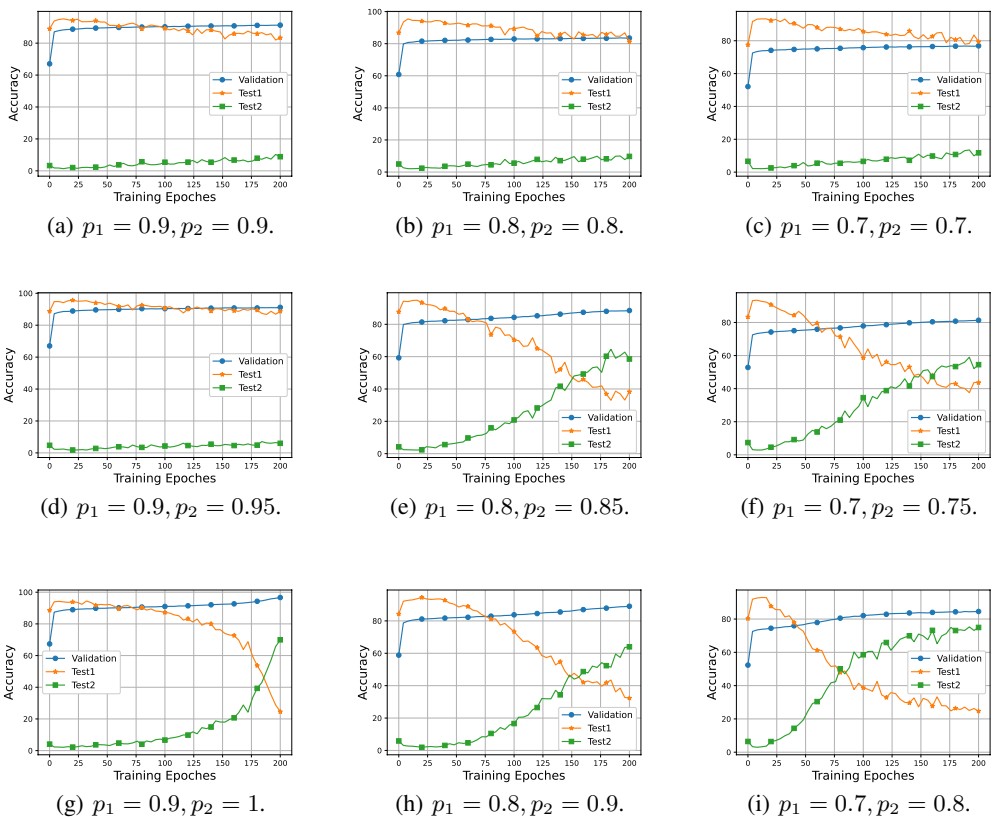

Figure 5: MLP' s performance with different $p_1, p_2$.

### B.3 PROOF OF THEOREM 2

**Theorem 5.** *An MoE module in equation 3 with $N$ experts and $k = 1$ aligns the conditional sentences in Algorithm 1 with*

$$
Alignment = \begin{cases} (N+1) \cdot \max \left( \mathcal{M}^*_{\mathscr{P}}, \mathcal{M}(G, h_1, \epsilon, \delta) \right), & \text{if } N < M, \\ (N+1) \cdot \max \left( \max_{i \in \{1, \cdots, M\}} \mathcal{M}(f_{FFN_i}, h_{i+1}, \epsilon, \delta), \mathcal{M}(G, h_1, \epsilon, \delta) \right), & \text{if } N \geq M, \end{cases}
$$

*where $\mathcal{M}(\cdot, \cdot, \cdot, \cdot)$ is defined in Definition 1, and $\mathcal{M}^*_{\mathscr{P}}$ is the optimal objective value of the following optimization problem:*

$$
\mathscr{P}: \underset{\mathcal{I}_1, \cdots \mathcal{I}_N}{minimize} \qquad \max_{i \in \{1, \cdots, N\}} \quad \mathcal{M}(f_{FFN_i}, ([1_{I_j}]_{j \in \mathcal{I}_i} \circ h_1)^T \cdot [h_j]_{j \in \mathcal{I}_i}, \epsilon, \delta)
$$

$$
subject \ to \qquad \cup_{i=1}^N \mathcal{I}_i = \{2, 3, \cdots, M+1\},
$$

*where $1_{I_i}$ is the indicator function on interval $I_i$.*

*Proof.* For $N \geq M$, we assign one expert for each function. For $N < M$, similar functions should be assigned to one expert to minimize the sample complexity.

**Case $N \geq M$:** We define the mapping $H(x) = [1_{I_1}, \cdots, 1_{I_M}] \in \mathbb{R}^M$, where $1_{I_i}$ is the indicator function on interval $I_i$. To show the alignment between MoE and conditional sentences, we define the module functions $f_1, \cdots, f_N$ and neural network modules $\mathcal{N}_1, \cdots, \mathcal{N}_N$ as

$$
f_i = \begin{cases} H \circ h_1, & \text{if } i = 1, \\ h_{i+1}, & \text{if } 1 \leq i \leq M+1, \\ 0, & \text{o.w.}, \end{cases} \qquad \mathcal{N}_i = \begin{cases} G, & \text{if } i = 1, \\ f_{FFN_i}, & \text{if } 1 \leq i \leq M+1, \\ f_{FFN_i}, & \text{o.w.}, \end{cases}
$$

Replacing $\mathcal{N}_i$ with $f_i$, the MoE network simulates Algorithm 1. Thus, for $N > M$, MoE algorithmically aligns conditional sentences with

$$
Alignment = (N+1) \cdot \max \left( \max_i \mathcal{M}(f_{FFN_i}, h_{i+1}, \epsilon, \delta), \mathcal{M}(G, h_1, \epsilon, \delta) \right) \tag{8}
$$

where $\mathcal{M}(\cdot, \cdot, \cdot, \cdot)$ is defined in Definition 1.

**Case $N < M$:** We define the mapping $H_0(x) = [1_{\cup_{j \in \mathcal{I}_1} I_j}, \cdots, 1_{\cup_{j \in \mathcal{I}_N} I_j}] \in \mathbb{R}^N$, where $\mathcal{I}_i$ is the solution to the optimization problem in equation 5. To show the alignment between MoE and conditional sentences, we define the module functions $f_1, \cdots, f_N$ and neural network modules $\mathcal{N}_1, \cdots, \mathcal{N}_N$ as

$$
f_i = \begin{cases} H_0 \circ h_1, & \text{if } i = 1, \\ ([1_{I_j}]_{j \in \mathcal{I}_{i-1}} \circ h_1)^T \cdot [h_j]_{j \in \mathcal{I}_{i-1}}, & \text{o.w.}, \end{cases} \qquad \mathcal{N}_i = \begin{cases} G, & \text{if } i = 1, \\ f_{FFN_i}, & \text{o.w.} \end{cases}
$$

Replacing $\mathcal{N}_i$ with $f_i$, the MoE network simulates Algorithm 1. Thus, for $N < M$, MoE algorithmically aligns with conditional sentences with

$$
Alignment = (N+1) \cdot \max \left( \mathcal{M}^*_{\mathscr{P}}, \mathcal{M}(G, h_1, \epsilon, \delta) \right)
$$

where $\mathcal{M}^*_{\mathscr{P}}$ is the optimal objective value to equation 5 and $\mathcal{M}(\cdot, \cdot, \cdot, \cdot)$ is defined in Definition 1. $\square$

### B.4 TECHNICAL LEMMAS

**Lemma 1.** *Under Assumption 1, for a given functions $f(\cdot)$ and an interval $A$, $\mathbb{P}_{D_{tr}}[f(\mathcal{N}_1(\boldsymbol{x})) \in A] < \delta$ implies $\mathbb{P}_{D_{te}}[f(\mathcal{N}_1(\boldsymbol{x})) \in A] \leq C\delta$.*

*Proof.* We denote the inverse mapping as $f^{-1}(A) = \{\boldsymbol{s} | f(\boldsymbol{s}) \in A\}$. Note that

$$
\mathbb{P}_{D_{te}}[f(\mathcal{N}_1(\boldsymbol{x})) \in A] = \int_{\boldsymbol{s} \in f^{-1}(A)} p_{\text{test}, \mathcal{N}_1}(\boldsymbol{s}) d\boldsymbol{s}
$$

$$
\leq C \int_{\boldsymbol{s} \in f^{-1}(A)} p_{\text{train}, \mathcal{N}_1}(\boldsymbol{s}) d\boldsymbol{s} = C\mathbb{P}_{D_{tr}}[f(\mathcal{N}_1(\boldsymbol{x})) \in A]
$$

This finishes the proof for Lemma 1. $\square$

**Remark 5.** Assumption 1 is similar to $\mu$-exploration in batch reinforcement learning (RL), which aims at controlling the distribution shift in RL (Xie & Jiang, 2021). Lemma 1 suggests that the generalization with the same training and test support resembles the IID generalization. Unfortunately, this is the largest distribution shift we can assume as the impossibility of nonlinear out-of-support generalization is proved in Xu et al. (2020b).

## C  IMPLEMENTATIONS

### C.1  DETAILED ARCHITECTURE

For image input $\mathbf{X}$, we first process the images as a sequence of equal-sized patches $\boldsymbol{x}_i \in \mathbf{X}$ using a 1-layer convolutional neural network with layer normalization. Then those patches are added with positional embeddings, and the patch embeddings (tokens) are ready to be processed by later $L$ blocks.

To make a fair comparison with other reported methods on DomainBed, we choose the ViT-S/16 which has similar parameters and run-time memory cost with ResNet-50 as the basic architecture for our main model GMoE-S/16. The chosen ViT-S/16 model has an input patch size of $16 \times 16$, 6 heads in multi-head attention layers, and 12 transformer blocks.

We consider two-layer configurations, *every-two* and *last-two*, that are widely adopted in Sparse MoE design (Riquelme et al., 2021). The detailed architectures are illustrated in Figure 6. Besides, we also consider evaluating GMoE's generalization performance with larger scale models (*e.g.,* ViT-Base).

V-MoE (Riquelme et al., 2021) trains 32 experts model on ImageNet-21K dataset. However, considering the number of training data, we need to adopt a smaller number of experts. More experts require larger datasets to ensure that every expert is adequately trained since each expert only sees a small portion of the dataset. Domain generalization datasets are usually 1-2 orders of magnitude smaller than ImageNet-21K. Based on this fact, each GMoE block contains 6 experts. For each image patch, the cosine router selects the TOP-2 index out of 6 and routes the patch to the corresponding 2 experts.

We put the ablation study results on layer configuration and larger size backbone model on Sec. D.6.

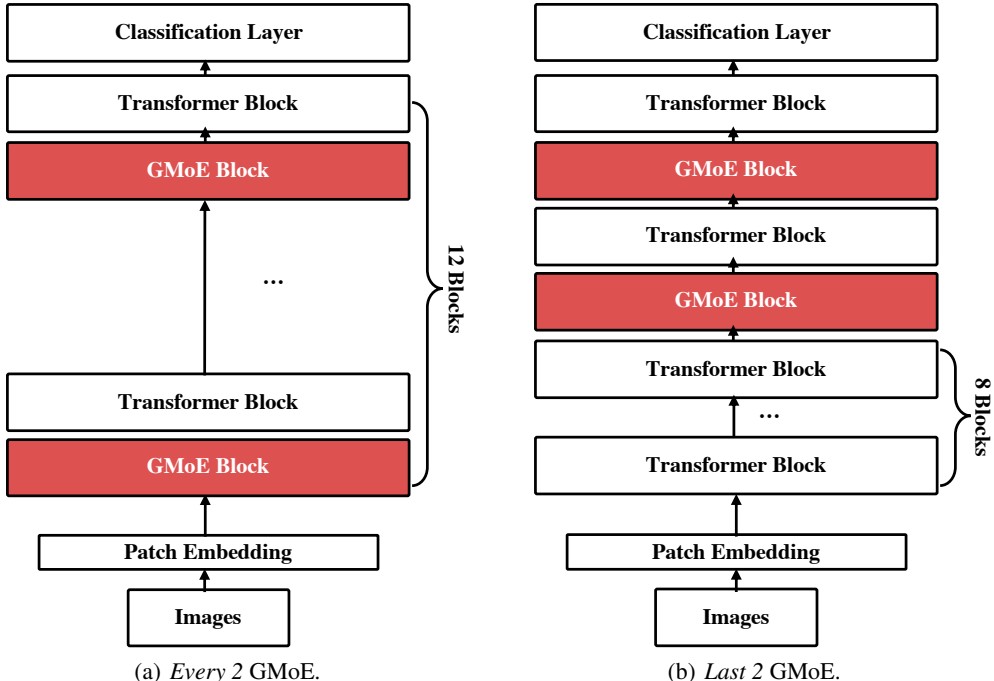

(a) *Every 2* GMoE.  (b) *Last 2* GMoE.

Figure 6: Diagrams of detailed GMoE architecture with two types of layer configuration.

## C.2  MODEL INITIALIZATION

To make a fair comparison with other algorithms from DomainBed (mostly with ResNet-50 pretrained on ImageNet-1K), we also utilize the pretrained ViT models on ImageNet-1K from DeiT to initialize our GMoE. Our practice avoids the generalization performance improvement of GMoE coming from a stronger pretrained model (*e.g.,* directly from V-MoE pretrained on ImageNet-21K). Meanwhile, there is no released pretrained MoE model on ImageNet-1K. In Table 4, we see ViT-S/16 and GMoE-S/16 achieve excellent generalization performance even with fewer parameters and smaller architecture.

In the GMoE block, we initialize the multiple experts with pretrained ViT block's FFN in the same position (*e.g.,* block-8 and 10) and multiply it by $E$ copies ($E$ is the number of experts). For cosine routers, we randomly initialize them in each GMoE block, to make them evenly choose experts at first.

Table 4: Comparison of the different backbone models used in the experiment. ImageNet-1K* denotes GMoE are initialized with ViT pretrained on ImageNet-1K and copy weights for different experts.

| Backbone | ResNet-50 | ResNet-101 | ViT-S/16 | GMoE-S/16 |
|---|---|---|---|---|
| **Pretrained Dataset** | ImageNet-1K | ImageNet-1K | ImageNet-1K | ImageNet-1K* |
| **Parameters** | 25.6M | 42.9M | 21.7M | 33.8M |
| **Flops** | 4.1G | 7.9G | 4.6G | 4.8G |

## C.3  PERTURBATION

Another specific design for the router is adding Gaussian noise to increase stability. Hard routing brings sparsity benefits while also causing discontinuities in routing. In a TOP-2 selection routing, a small perturbation would cause order changes if the largest output and second-largest output are close. In domain generalization, data come from diverse domains and we need to learn a model to capture the invariant semantic meaning of an object. Existing work (Chen et al., 2022) theoretically claims that adding noise to the gating function provides a smooth transition of the gating function during training. Empirically, we find adding noise with a standard deviation of $\frac{1}{N}$ would empirically improve performance and stabilize training dynamics.

## C.4  AUXILIARY LOSS

It has been observed that the gating network tends to converge to a self-reinforcing imbalanced state, where a few experts are more frequently activated than others. Following Riquelme et al. (2021); Shazeer et al. (2017), we define an importance loss $\mathcal{L}_{\text{imp}}$ to encourage balanced usage of experts. The importance of expert $e$ is defined as the normalization of the gating function's output in a batch of images:

$$\text{imp}_e(\mathbf{X}) = \sum_{\mathbf{x} \in \mathbf{X}} g(\boldsymbol{x})_e \tag{9}$$

And the loss $\mathcal{L}_{imp}$ is defined by the squared coefficient of variation of $\text{imp}(\mathbf{X})$:

$$\mathcal{L}_{\text{imp}} = \left( \frac{\text{STD}(\text{imp}(\mathbf{X}))}{\text{MEAN}(\text{imp}(\mathbf{X}))} \right)^2$$

While importance loss seeks that all experts have on average similar output routing weights, it is still likely some experts would not be routed (see example in Table 5).

To address this issue, besides balancing across patches, we also need to encourage balanced assignment across experts. Ideally, we expect a balanced assignment via a load loss $\mathcal{L}_{\text{load}}$. However, the assignment is a discrete value; then we expect a proxy to make it differentiable in back-propagation. Following the idea in Riquelme et al. (2021); Shazeer et al. (2017); Mustafa et al. (2022), for each token $\mathbf{x}$ and each expert $e_i$, we could compute the probability of $e_i$ once being selected in $\text{TOP}_k$

Table 5: Simple example on $\text{TOP}_1$ score (before SoftMax operation) where average weights across patches ($x_{1,2,3}$) are balanced. However, Expert 2 has always been ignored.

| Patch | Expert 1 | Expert 2 | Expert 3 | Expert 4 | Selected Expert |
|:-----:|:--------:|:--------:|:--------:|:--------:|:---------------:|
| $x_1$ | 0.9 | 0.4 | 0.1 | 0.2 | Expert 1 |
| $x_2$ | 0.2 | 0.4 | 0.9 | 0.1 | Expert 3 |
| $x_3$ | 0.1 | 0.4 | 0.2 | 0.9 | Expert 4 |

after SoftMax operation, and then kept if re-sampling again *only* the noise for expert $e_i$. In other words, we are computing the probability that having expert $e_i$ still being among the $\text{TOP}_k$ while re-sampling again *only* the noise of the expert. This probability is formally defined as in Eq. 10, where $\eta_k$ the $K$-th largest entry after SoftMax operation and $\Phi$ is the cumulative distribution function of a Gaussian distribution.

$$p_e(\boldsymbol{x}) = 1 - \Phi\left(\frac{\eta_k - (\boldsymbol{E}^T\boldsymbol{W}\boldsymbol{x})_e}{\sigma}\right) \tag{10}$$

Then the expert load could be defined as $\text{load}_e(\mathbf{X}) = \sum_{x \in \mathbf{X}} p_e(\boldsymbol{x})$ and the whole experts load is defined as $\text{load}(\mathbf{X}) = \{\text{load}_e(\mathbf{X})\}_{e=1}^{E}$ .The load loss $\mathcal{L}_{\text{load}}$ is defined by

$$\mathcal{L}_{\text{load}}(\mathbf{X}) = \left(\frac{\text{STD}(\text{load}(\boldsymbol{x}))}{\text{MEAN}(\text{load}(\boldsymbol{x}))}\right)^2$$

With the above adaptations for DG tasks, GMoE still works with a simple overall loss:

$$\mathcal{L}(\mathbf{X}) = \mathcal{L}_{\text{classification}}(\mathbf{X}) + \frac{1}{2}\lambda(\mathcal{L}_{\text{imp}}(\mathbf{X}) + \mathcal{L}_{\text{load}}(\mathbf{X}))$$

with a hyperparameter $\lambda > 0$ to balance the main task loss and the auxiliary loss, in which usually we view equal importance of the load and importance loss.

# D EXPERIMENTAL RESULTS

## D.1 DOMAINBED DETAILS

**Benchmark Datasets** Following previous DG studies, we mainly evaluate our proposed method and report empirical results on DomainBed (Gulrajani & Lopez-Paz, 2021) with 8 benchmark datasets, PACS, VLCS, OfficeHome, TerraIncognita, DomainNet, SVIRO, Wilds-Camelyon and Wilds-FMOW. Besides DomainBed, we also conduct experiments on the self-created CUB-DG dataset for model analysis. In the following, we will provide the details of different datasets.

**Dataset Details** In Table 6, we provide statistics for the 8 datasets in DomainBed, as well as our self-created dataset CUB-DG (e.g., number of domains, categories and images).

Table 6: Statistics of DomainBed datasets.

| Dataset | PACS | VLCS | OfficeHome | TerraInc | DomainNet |
|---|---|---|---|---|---|
| # Domains | 4 | 4 | 4 | 4 | 6 |
| # Classes | 7 | 5 | 65 | 10 | 345 |
| # Examples | 9,991 | 10,729 | 15,588 | 24,788 | 586,575 |

Table 7: Statistics of Wilds and CUB-DG. TerraInc stands for TerraIncognita, Camelyon stands for Wilds-Camelyon and FMOW stands for Wilds-FMOW.

| Dataset | SVIRO | W-Camelyon | W-FMOW | CUB-DG |
|---|---|---|---|---|
| # Domains | 10 | 5 | 6 | 4 |
| # Classes | 7 | 2 | 62 | 200 |
| # Examples | 56,000 | 455,954 | 523,846 | 47,152 |

In detail, the 9 multi-domain image classification datasets are comprised of:

1. **PACS** (Li et al., 2017) comprises four domains $d \in \{\text{art, cartoons, photos, sketches}\}$. This dataset contains $9,991$ examples of dimension $(3, 224, 224)$ and 7 classes.

2. **VLCS** (Fang et al., 2013) comprises photographic domains $d \in \{\text{Caltech101, LabelMe, SUN09, VOC2007}\}$. This dataset contains $10,729$ examples of dimension $(3, 224, 224)$ and 5 classes.

3. **Office-Home** (Venkateswara et al., 2017) includes domains $d \in \{\text{art, clipart, product, real}\}$. This dataset contains $15,588$ examples of dimension $(3, 224, 224)$ and 65 classes.

4. **TerraIncognita** (Beery et al., 2018) contains photographs of wild animals taken by camera traps at locations $d \in \{\text{L100, L38, L43, L46}\}$. Our version of this dataset contains $24,788$ examples of dimension $(3, 224, 224)$ and 10 classes.

5. **DomainNet** (Peng et al., 2019) has 6 domains $d \in \{\text{clipart, infograph, painting, quickdraw, real, sketch}\}$. This dataset contains $586,575$ examples of size $(3, 224, 224)$ and 345 classes.

6. **SVIRO** (Cruz et al., 2020) is a Synthetic dataset for Vehicle Interior Rear seat Occupancy across 10 different vehicles, the 10 domains.
   The domains are $d \in \{\text{aclass, escape, hilux, i3, lexus, tesla, tiguan, tucson, x5, zoe}\}$. This dataset for image classification contains $56,000$ image examples of size $(3, 224, 224)$ and 7 classes.

7. **Wilds-Camelyon** (Koh et al., 2021) dataset contains histopathological image slides collected and processed by different hospitals and curated by Wilds benchmark. It contains $455,954$ examples of dimension $(3, 224, 224)$ and 2 classes from 5 hospitals.

8. **Wilds-FMOW** (Koh et al., 2021) dataset is a variant of the functional map of the world dataset (Christie et al., 2018) and contains satellite images of 62 buildings or land classes across 6 regions. The image example is of size $(3, 224, 224)$.

Table 8: Hyperparameters to reproduce best performance of GMoE on each dataset.

| Hyperparameters | PACS | VLCS | OfficeHome | TerraInc | DomainNet |
|---|---|---|---|---|---|
| Learning Rate | $3 \times 10^{-5}$ | $3 \times 10^{-5}$ | $1 \times 10^{-5}$ | $5 \times 10^{-5}$ | $5 \times 10^{-5}$ |
| Weight Decay | 0 | $1 \times 10^{-6}$ | $1 \times 10^{-6}$ | $1 \times 10^{-4}$ | 0 |

9. **CUB-DG** dataset is built from the original Caltech-UCSD Birds (CUB) dataset (Wah et al., 2011). We stylize the original images into three domains, Candy, Mosaic, and Undie, while keeping the original images as the Natural domain. Overall, CUB-DG has 4 domains $d \in$ {Natural, Candy, Mosaic, Udnie}. It contains 47,152 bird examples and are categorized into 200 classes. Each image has detailed annotations: 1 category label, 15 positional attributes (*e.g.,* left wing, back), 312 binary attributes (*e.g.,* `has_bill_shape=dagger`), and 1 bounding box information. We treat the 15 positional attributes as visual attributes of a bird image, and thus we could evaluate the correlation between visual attributes and expert selections.

**Evaluation protocols**    We evaluate our proposed method with two protocols on DomainBed (Gulrajani & Lopez-Paz, 2021) benchmark.

For **train-validation selection**, we split each training domain into training and validation subsets. Then, we pool the validation subsets of each training domain to create an overall validation set. Finally, we choose the model maximizing the accuracy on the overall validation set, and report the final accuracy on one leave-out test domain.

For **leave-one-domain-out validation selection**, we train the model on all training domains and one domain out as the validation set. We choose the model maximizing the accuracy on the leave-out validation domain, and report accuracy on another leave-out test domain. We should emphasize that the leave-one-domain-out validation setting means we choose two domains as leave-out domains. This term is used in conformity in literature (Cha et al., 2022; 2021b).

**Standard error bars**    We train the model three times with random seeds on weight initializations. The mean and standard error of these repetitions are reported.

### D.2    TRAINING DETAILS & HYPERPARAMETERS

In this section, we report the training details of our experiments. We optimize models using Adam optimizer (Kingma & Ba, 2015) with slightly different parameters on different datasets (see Table 8). The training and inference batch size is set to 32 for each domain.

Since Gulrajani & Lopez-Paz (2021) only evaluates baselines for 5K iterations on DomainNet, while 5K iterations are less than 2 epochs on DomainNet. This results in insufficient training on this dataset and it can not fully reveal models' performance with different algorithms. Subsequent state-of-the-art works (Cha et al., 2021b) (Cha et al., 2022) re-evaluated ERM and proposed their methods on 15,000 iterations of DomainNet.

To compare with those SOTA counterparts, we report the results of 15K iterations on DomainNet in Table 1. However, we also train GMoE for 5K iterations to make a fair comparison with both types of algorithms (see Table 9). For the rest of the datasets on DomainBed, we train GMoE for 5K iterations following the original setting.

### D.3    LEAVE-ONE-DOMAIN-OUT RESULTS

We have demonstrated the experimental results in Table 1 with train-validation selection. In this section, we will further provide the results of GMoE on leave-one-domain-out selection, which is a more challenging setting. In this setting, we should train GMoE on 5 datasets with in total of 61 individual experiments and summarize them as average performance. Due to the huge amount of experiments, recent methods rarely evaluate their methods on such setting, except for those provided

Table 9: Train-validation selection performance comparison with 5K and 15K training iterations, where *train-val* stands for train-validation selection and *lodo* stands for leave-one-domain-out selection. The place marked — is not applicable as stated in Sec. D.3.

| Iterations | Algorithms & Models | DomainNet (*train-val*) | DomainNet (*lodo*) |
|---|---|---|---|
| 5K | ERM (Vapnik, 1991) | 41.2 ± 0.2 | 40.6 ± 0.2 |
| | IRM (Arjovsky et al., 2019) | 33.9 ± 2.8 | 33.5 ± 3.0 |
| | DANN (Ganin et al., 2016) | 41.8 ± 0.2 | 38.2 ± 0.2 |
| | CORAL (Sun & Saenko, 2016) | 41.8 ± 0.2 | 41.1 ± 0.1 |
| | MMD (Li et al., 2018b) | 39.4 ± 0.8 | 23.4 ± 9.4 |
| | MLDG (Li et al., 2018a) | 41.2 ± 0.1 | 41.0 ± 0.2 |
| | Fish (Shi et al., 2021) | 42.7 ± 0.2 | — |
| | Fishr (Rame et al., 2021) | 41.7 ± 0.2 | — |
| | ViT-S/16 (Dosovitskiy et al., 2021) | 42.4 ± 0.0 | 42.0 ± 0.2 |
| | **GMoE-S/16** | **44.6 ± 0.3** | **44.7 ± 0.2** |
| 15K | Swad (Cha et al., 2021a) | 46.5 ± 0.1 | — |
| | MIRO (Cha et al., 2022) | 47.4 ± 0.0 | — |
| | ViT-S/16 (Dosovitskiy et al., 2021) | 47.1 ± 0.3 | 46.1 ± 0.4 |
| | **GMoE-S/16** | **48.7 ± 0.2** | **48.4 ± 0.3** |

by DomainBed. In the Table 10, we can find that GMoE outperforms previous methods in all 5 datasets.

Table 10: Overall out-of-domain accuracies (%) with leave-one-domain-out selection criterion.

| Algorithm | PACS | VLCS | OfficeHome | TerraInc | DomainNet |
|---|---|---|---|---|---|
| ERM (ResNet50) | 83.0 ± 0.7 | 77.2 ± 0.4 | 65.7 ± 0.5 | 41.4 ± 1.4 | 40.6 ± 0.2 |
| IRM | 81.5 ± 0.8 | 76.3 ± 0.6 | 64.3 ± 1.5 | 41.2 ± 3.6 | 33.5 ± 3.0 |
| DANN | 81.0 ± 1.1 | 76.9 ± 0.4 | 64.9 ± 1.2 | 44.4 ± 1.1 | 38.2 ± 0.2 |
| CORAL | 82.6 ± 0.5 | 78.7 ± 0.4 | 68.5 ± 0.2 | 46.3 ± 1.7 | 41.1 ± 0.1 |
| MMD | 83.2 ± 0.2 | 77.3 ± 0.5 | 60.2 ± 5.2 | 46.5 ± 1.5 | 23.4 ± 9.5 |
| MLDG | 77.2 ± 0.9 | 82.9 ± 1.7 | 66.1 ± 0.5 | 46.2 ± 0.9 | 41.0 ± 0.2 |
| ViT-S/16 | 86.6 ± 0.1 | 78.3 ± 0.2 | 71.9 ± 0.2 | 41.9 ± 0.3 | 46.1 ± 0.2 |
| **GMoE-S/16** | **87.1 ± 0.3** | **80.0 ± 0.1** | **73.9 ± 0.2** | **46.7 ± 0.3** | **48.4 ± 0.2** |

## D.4 ABLATION STUDY WITH DG ALGORITHMS

This section presents ablation study to compare GMoE with DG algorithms and ViT with DG algorithms, as shown in Table 11. From the table, we see that GMoE significantly outperforms ViT and ResNet under a variety of DG algorithms, showing the generalizability of the proposed backbone architecture.

## D.5 SINGLE-SOURCE DOMAIN GENERALIZATION RESULTS

In this section, we will demonstrate the *single source domain generalization* results on DomainNet's 6 domains. We select different training domains in turn and take the remaining 5 domains as test domains. In Table 12, we show the OOD results when models are trained on *only one* domain and test on 5 test domains, as well as the IID results on the training domain's validation set. We specify IID and OOD results in different colors.

Table 11: Overall accuracies (%) with train-validation selection criterion.

| Algorithms | Backbones | PACS | VLCS | OfficeHome | TerraInc | DomainNet | Avg |
|---|---|---|---|---|---|---|---|
| Swad | ViT-S/16 | 88.3 | 80.4 | 72.7 | 46.9 | 48.8 | 67.4 |
| | **GMoE-S/16** | **88.6** | **80.8** | **74.5** | **49.1** | **49.6** | **68.5** |
| Fish | ResNet50 | 85.5 | 77.8 | 68.6 | 45.1 | 43.1 | 64 |
| | ViT-S/16 | 87.3 | 78 | 72.2 | 42.7 | 47.3 | 65.5 |
| | **GMoE-S/16** | **87.9** | **79.7** | **73.0** | **46.9** | **48.8** | **67.1** |
| Fishr | ResNet50 | 85.5 | 77.8 | 68.6 | 47.4 | 41.7 | 64.2 |
| | ViT-S/16 | 85.6 | **79.1** | 71.8 | 44.1 | 47.6 | 65.6 |
| | **GMoE-S/16** | **88.3** | 78.8 | **72.7** | **45.6** | **48.3** | **66.7** |

## D.6 ABLATION STUDY ON MODEL DESIGN

In order to validate discussions in Sec. 4.3 and Sec C.1, we conduct experiments with different layer configurations, as well as adopting larger backbone (*e.g.,* ViT-Base) model. From the results in Table 13, we see that the *last two* configuration largely exceeds *every two* configuration, thus verifying our idea that visual attributes exist in relatively high-level signals. This result also demonstrates that the cosine router and larger model lead to better generalization performance.

## D.7 ABLATION STUDY ON RESNET WITH STRONG DATA AUGMENTATION

In this part, we will go into greater depth about the pre-trained model's effect on DG performance. As stated in Sec. C.2, GMoE is initialized from the same architecture ViT models, where pre-trained weights are from Touvron et al. (2021). In addition, the pre-training on ViT and ResNet models may adopt different training recipes. And it is generally believed that models with stronger data augmentation during pre-training could have better performance on downstream tasks.

To verify this difference in detail, we consider comparing GMoE-S/16's pre-trained model, ViT-S/16, with a stronger pre-trained ResNet-50 V2 model[1] with significant training tricks data augmentations. The pre-training details on ViT-S/16 and ResNet V2 are listed in Table 14.

Due to the space limit, we use abbreviations for some terms. In detail, they are:

- **LR Opt.** stands for learning rate optimization.
- **TrivialAug.** stands for trivial augmentation (Müller & Hutter, 2021).
- **Ep.** stands for pre-training epochs.
- **Rand Er.** stands for random erasing (Zhong et al., 2017).
- **Label Sm.** stands for label smoothing (Szegedy et al., 2016).
- **FixRes Mt.** stands for FixRes mitigations (Touvron et al., 2019).
- **WDT** stands for weight decay tuning.
- **IRT** stands for inference resize tuning (Touvron et al., 2019).

The results in Table 15 demonstrate that GMoE and ViT could still remain higher performance than the strong pre-trained model ResNet V2 due to the benefits of the backbone architecture.

## D.8 COMPUTATIONAL COST COMPARISON

Having shown the effects of GMoE's performance on different domain generalization benchmarks, we now conduct efficiency analysis with respect to **iteration time** and **run-time memory** during training and inference. Since algorithms developed from DomainBed mainly adopt the same architectures

---

[1]From TORCHVISION's Pretrained Models.

Table 12: Single-source DG results on DomainNet. We alternatively choose 1 training domain and the other 5 as test domains. We report  IID generalization  results on training domain's validation set with grey color, and  OOD generalization  results on test domains' validation set with light cyan color.

| Clipart | Clipart | Info | Paint | Quick | Real | Sketch |
|---|---|---|---|---|---|---|
| ResNet50 | 68.5 | 12.3 | 25.8 | 9.5 | 39.6 | 37.1 |
| ResNet101 | 70.4 | 12.2 | 28.6 | 12.4 | 43.0 | 40.7 |
| ViT-S/16 | 74.7 | 16.2 | 35.3 | 12.7 | 50.9 | 40.4 |
| GMoE-S/16 | **76.3** | **16.8** | **36.2** | **13.9** | **51.6** | **42.6** |
| **Info** | Clipart | Info | Paint | Quick | Real | Sketch |
| ResNet50 | 27.9 | 34.0 | 23.3 | 2.0 | 34.3 | 24.5 |
| ResNet101 | 29.2 | 35.6 | 24.0 | 3.1 | 34.1 | 24.6 |
| ViT-S/16 | 36.3 | 40.2 | 33.6 | **5.1** | 47.4 | 30.7 |
| GMoE-S/16 | **36.7** | **41.0** | **34.2** | 4.5 | **48.2** | **30.8** |
| **Paint** | Clipart | Info | Paint | Quick | Real | Sketch |
| ResNet50 | 37.1 | 12.9 | 62.7 | 2.2 | 49.3 | 33.3 |
| ResNet101 | 40.5 | 13.1 | 63.4 | 3.1 | 51.2 | 35.4 |
| ViT-S/16 | 42.7 | 15.9 | 69.0 | 5.0 | **56.4** | 37.0 |
| GMoE-S/16 | **43.5** | **16.1** | **69.3** | **5.3** | **56.4** | **38.0** |
| **Quick** | Clipart | Info | Paint | Quick | Real | Sketch |
| ResNet50 | 17.3 | 1.1 | 2.8 | 62.4 | 7.3 | 9.2 |
| ResNet101 | 16.0 | 1.3 | 2.7 | 64.1 | 6.3 | 8.5 |
| ViT-S/16 | 22.1 | **2.0** | 6.3 | 66.2 | 11.8 | 11.8 |
| GMoE-S/16 | **22.8** | **2.0** | **6.4** | **66.8** | **12.2** | **12.9** |
| **Real** | Clipart | Info | Paint | Quick | Real | Sketch |
| ResNet50 | 39.5 | 13.0 | 40.0 | 4.4 | 73.6 | 30.5 |
| ResNet101 | 43.3 | 15.6 | 43.3 | 6.8 | 75.3 | 34.0 |
| ViT-S/16 | 46.9 | 18.0 | 46.0 | 5.0 | 79.5 | 33.4 |
| GMoE-S/16 | **47.4** | **18.5** | **46.9** | **7.5** | **79.6** | **36.4** |
| **Sketch** | Clipart | Info | Paint | Quick | Real | Sketch |
| ResNet50 | 48.9 | 11.7 | 31.6 | 10.8 | 40.7 | 63.4 |
| ResNet101 | 48.4 | 11.8 | 31.9 | 11.3 | 38.1 | 64.3 |
| ViT-S/16 | 52.4 | **14.4** | **38.6** | 13.4 | 48.2 | 67.4 |
| GMoE-S/16 | **52.9** | 14.1 | 36.6 | **13.7** | **49.0** | **68.4** |

(*e.g.,* ResNet50, Linear Classifier), traditional model complexity measures such as flops and model parameters can not truly reflect the difference in efficiency. By evaluating the above two metrics, different algorithms with more complex loss design or gradient constraints will cause larger overheads with respect to iteration time or run-time memory. From the results in Table 16, we observe that GMoE achieves relatively low run-time memory and training step time among other competitors.

Table 13: Train-validation selection performance comparison for different GMoE models. All GMoEs (S/16 and B/16) have $E = 6$ experts and $L = 12$ blocks. We specify the number of attention heads ($H$), the patch embedding size ($D$), layer configuration, and router's type in the table header.

| Algorithm | Config. | Router | $H$ | $D$ | PACS | VLCS | OfficeHome | TerraInc | DomainNet |
|-----------|---------|--------|-----|-----|------|------|-----------|----------|-----------|
| GMoE-S/16 | *Every 2* | Linear | 6 | 384 | 81.8 ± 0.2 | 75.0 ± 0.1 | 64.0 ± 0.4 | 32.5 ± 0.7 | 46.3 ± 0.3 |
| GMoE-S/16 | *Every 2* | Cosine | 6 | 384 | 81.4 ± 0.1 | 74.8 ± 0.2 | 62.2 ± 0.4 | 40.9 ± 0.3 | 46.4 ± 0.2 |
| GMoE-S/16 | *Last 2* | Linear | 6 | 384 | 87.8 ± 0.2 | 80.0 ± 0.0 | 72.7 ± 0.2 | 46.7 ± 0.2 | 48.3 ± 0.1 |
| GMoE-S/16 | *Last 2* | Cosine | 6 | 384 | 88.1 ± 0.1 | 80.2 ± 0.2 | 74.2 ± 0.4 | 48.5 ± 0.4 | 48.7 ± 0.2 |
| GMoE-B/16 | *Last 2* | Cosine | 12 | 768 | 89.4 ± 0.1 | 81.2 ± 0.1 | 77.2 ± 0.4 | 49.3 ± 0.3 | 51.3 ± 0.1 |

Table 14: Comparison of different training recipes on ResNet-50 V2 and ViT-S/16.

| Recipe | LR Opt. | TrivialAug. | Ep. | Rand Er. | Label Sm. | FixRes Mt. | WDT | IRT | IN1K |
|--------|---------|-------------|-----|----------|-----------|-----------|-----|-----|------|
| ResNet-50 V2 | ✔ | ✔ | 600 | ✔ | ✔ | ✔ | ✔ | ✔ | 80.8 |
| ViT-S/16 | ✔ | ✗ | 300 | ✔ | ✗ | ✗ | ✗ | ✔ | 79.9 |

Table 15: Train-validation selection performance comparison for ViT, GMoE, and other DG algorithms with ResNet-50 V2 as backbone model. On DomainNet, results are reported with 15K iterations.

| Algorithm | PACS | VLCS | OfficeHome | TerraInc | DomainNet |
|-----------|------|------|-----------|----------|-----------|
| ERM (w/ ResNet-50 V2) | 87.2 | 78.2 | 68.7 | 49.9 | 45.3 |
| Fishr (w/ ResNet-50 V2) | 87.5 | 77.9 | 70.4 | **51.7** | 47.0 |
| ViT-S/16 | 86.2 | 79.7 | 72.2 | 42.0 | 47.1 |
| **GMoE-S/16** | **88.1** | **80.1** | **74.2** | 48.5 | **48.7** |

Table 16: Comparison of training/inference iteration time and run-time memory for a mini-batch. A mini-batch is formed with 160 images in $224 \times 224$ resolutions from DomainBed. For both metrics, lower is better.

| Training | ERM | DANN | IRM | Fish | Fishr | SWAD | VIT-S/16 | GMoE-S/16 |
|----------|-----|------|-----|------|-------|------|----------|-----------|
| Step Time (s) ↓ | 1.01 | 1.02 | 1.10 | 2.79 | 1.10 | 1.21 | 0.90 | 0.98 |
| Run-time Memory (GB) ↓ | 13.40 | 13.42 | 13.40 | 3.41 | 15.25 | 14.32 | 11.15 | 12.28 |
| **Inference** | **ERM** | **DANN** | **IRM** | **Fish** | **Fishr** | **SWAD** | **VIT-S/16** | **GMoE-S/16** |
| Step Time (s) ↓ | 0.32 | 0.33 | 0.32 | 0.33 | 0.34 | 0.33 | 0.28 | 0.30 |
| Run-time Memory (GB) ↓ | 1.82 | 1.83 | 1.82 | 1.82 | 1.83 | 1.84 | 0.76 | 1.05 |

# E   MODEL ANALYSIS

## E.1   CUB-DG RESULTS & VISUALIZATIONS

**Generalization across Image Stylization**    To evaluate the model performance in generalizing to image stylization, we conduct experiments following DomainBed setting on CUB-DG. We compare GMoE with other three invariant learning DG algorithms, (i.e., DANN (Ganin et al., 2016), Fish (Shi et al., 2021) and Fishr (Rame et al., 2021)). Among them, DANN and Fish are widely discussed and cited papers, and Fishr is considered the best algorithm of its kind so far.

In this experiment, all three invariant learning methods, including ERM, adopt ResNet-50 as the backbone model. GMoE does not have any specific loss design and constraint. The only difference is the model architecture. In Table 17, we see that GMoE, the new backbone model for DG, is significantly more effective than those methods specifically designed to learn invariance with certain losses and gradient constraints.

Table 17: Out-of-domain accuracy (%) in each domain on CUB-DG dataset. The best is in bold.

| Algorithm | Candy | Mosaic | Natural | Udnie |
|---|---|---|---|---|
| ERM (ResNet50) | 64.3 | 20.8 | 75.3 | 77.9 |
| DANN | 47.0 | 14.9 | 76.5 | 70.6 |
| Fish | 62.6 | 22.4 | 84.5 | 77.7 |
| Fishr | 62.4 | 24.3 | 78.8 | 73.8 |
| **GMoE-S/16** | **83.1** | **42.8** | **89.3** | **82.7** |

**Experts Selection**    Following the analysis in Sec. 5.4, we provide more visualization results in this section.

In Figure 8-9, we provide the expert selection results in natural domain images across different bird classes. We use different colored lines that correspond to different visual attributes, and we connect these lines across images in a row. It indicates that experts focus on the same visual attributes while seeing different bird images from different angles. This suggests that mixture-of-experts have the ability to capture and process visual attributes *across class*.

In Figure 10-13, we provide the expert selection results for the same image on four domains (with four types of stylization). In order to more vividly display the expert selection across domains, we assign different colors to different selected experts. We see that, in most cases, the same areas and visual attributes are handled by the same experts across domains. For example, in Figure 10, expert 1,3 and 4 focus on the bird while expert 0,2 focus on the background. And it is consistent across four domains with different stylization. The results reveal that experts are potentially invariant in dealing with visual attributes *across domains*.

**Multi-head Attention Visualization**    Ideally, the multi-head attention mechanism can be viewed as jointly attending to multiple places by ensembling multiple attention heads. Each attention head would focus on its specific attention relationship between all patches and inherently collaborate with each other.

However, recent studies (Cordonnier et al., 2020) demonstrate that multiple-head attention (MHA) layers could learn redundant key/query projections, i.e. some heads might attend to similar features in input space. This issue demonstrates that two heads $\text{head}_i$ and $\text{head}_j$ are computing the same key/query representations up to a unitary matrix $\mathbf{I} \in \mathbb{R}^{d \times d}$ such that $W_{Q_i} = W_{Q_j}\mathbf{I}$ and $W_{K_i} = W_{K_j}\mathbf{I}$. In this case, even though the two heads are computing identical attention scores, i.e. $W_{Q_i}\mathbf{I}\mathbf{I}^T W_{K_i}^T$, the concatenation $[W_{Q_i}, W_{Q_j}] \in \mathbb{R}^{d \times 2d}$ can also be full rank, which indicates that some attention heads would focus on the same content but are agnostic to each other. We opine that the mixture-of-experts (MoE) layer could alleviate redundancy and increase disentanglement to some extent.

In Figure 7, presented in a grid fashion, the MHA layer focuses on different signals (features) in different heads while MoE layer disentangles the information by handling each patch to different

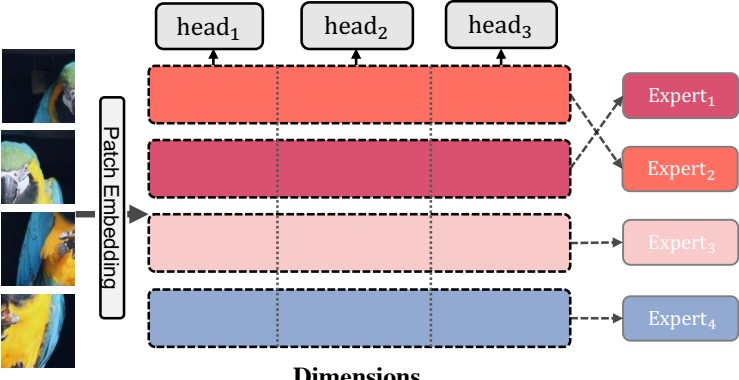

Figure 7: Diagram of a GMoE block, where different attention heads attend to different aspects of relations between patches. Different experts handle the learned attention in different patches. The router has been omitted for brevity.

experts. This mechanism extends the network sparsity and hence is able to improve the model generalizability. In sum, the MoE layer is designed to leverage different experts to capture distinct visual attributes, as well as their correlations (via attention mechanism) to each other.

To see more about the learned representation in multi-head attention layer, we provide the visualization of last layer's attention activations of ViT and GMoE on Figure 14 15 16 17.

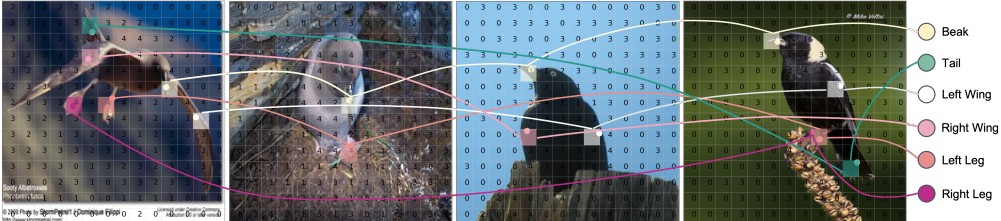

Figure 8: Expert selection visualization on block-10 of GMoE-S/16. Images are from different classes of natural domains on CUB-DG. Different colored lines connect the same type of visual attributes across images. Same visual attribute is processed by the same expert, *e.g.,* beak and tail by expert 3, left/right leg by expert 4.

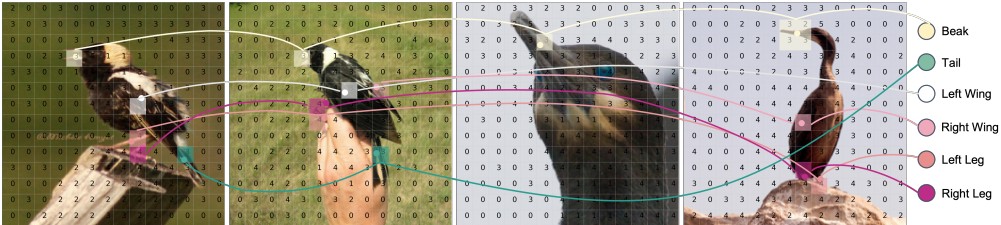

Figure 9: Expert selection visualization on block-10 of GMoE-S/16. Images are from different classes of natural domains on CUB-DG. Different colored lines connect the same type of visual attributes across images. Same visual attribute is processed by the same expert, *e.g.,* beak and tail by expert 3, left/right leg by expert 4.

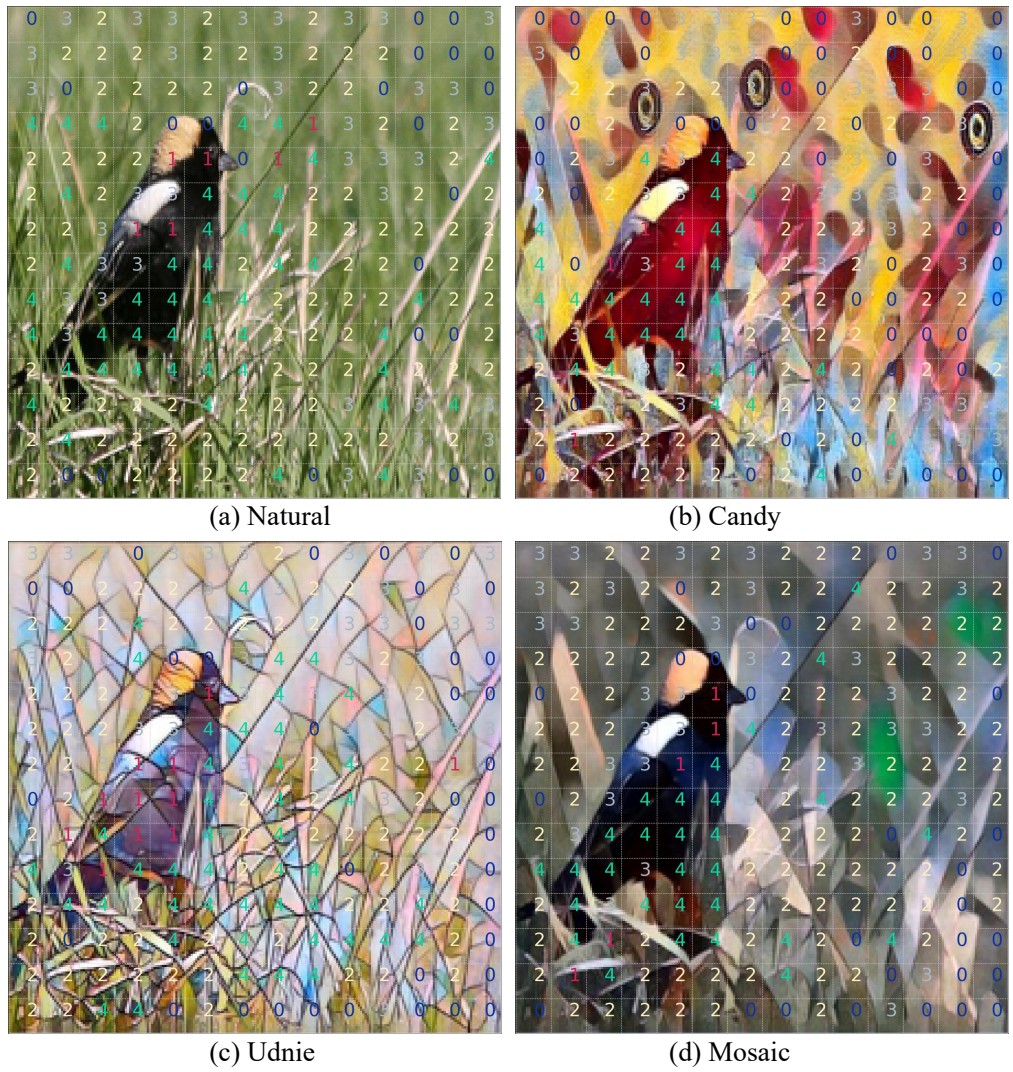

Figure 10: Expert selection visualization on block-10 of GMoE-S/16. Images are from different domains on CUB-DG.

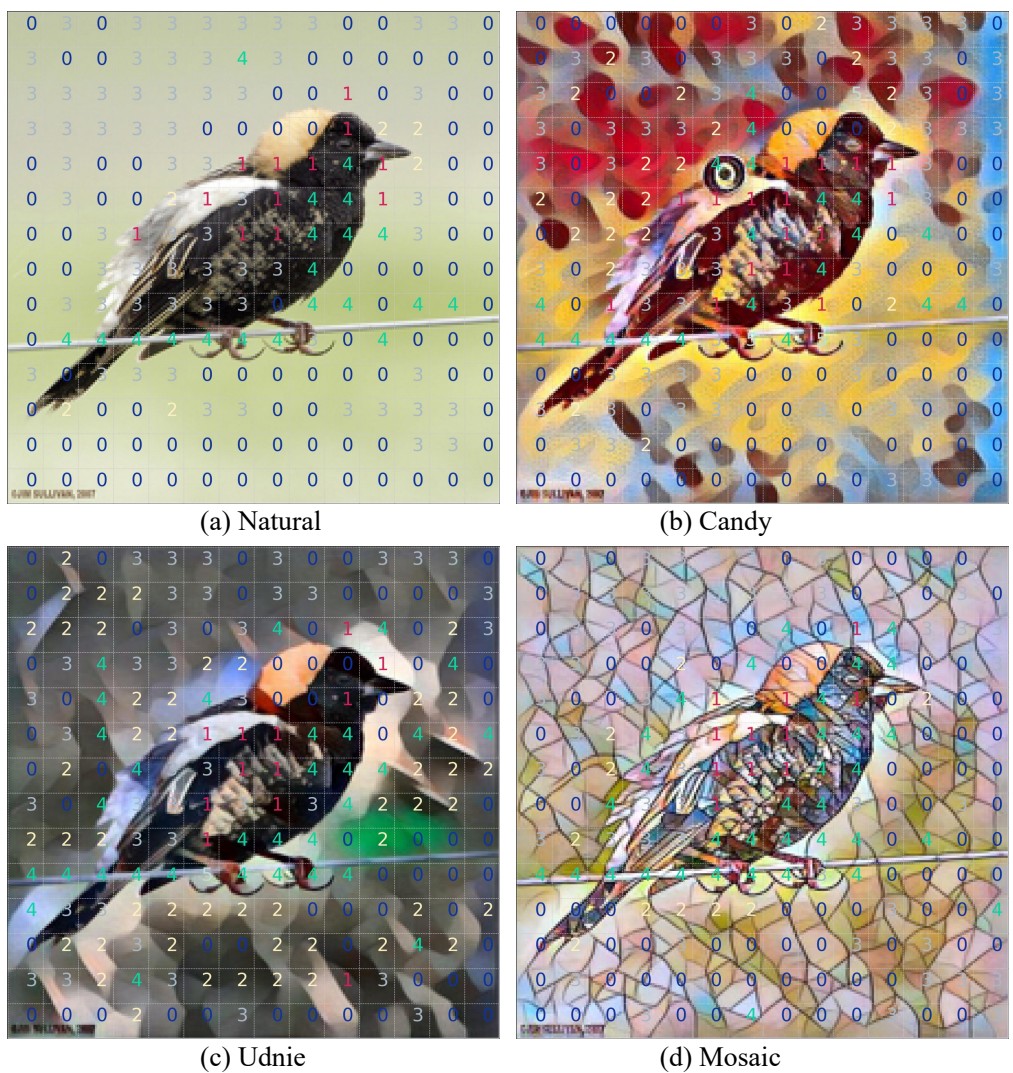

Figure 11: Expert selection visualization on block-10 of GMoE-S/16. Images are from different domains on CUB-DG.

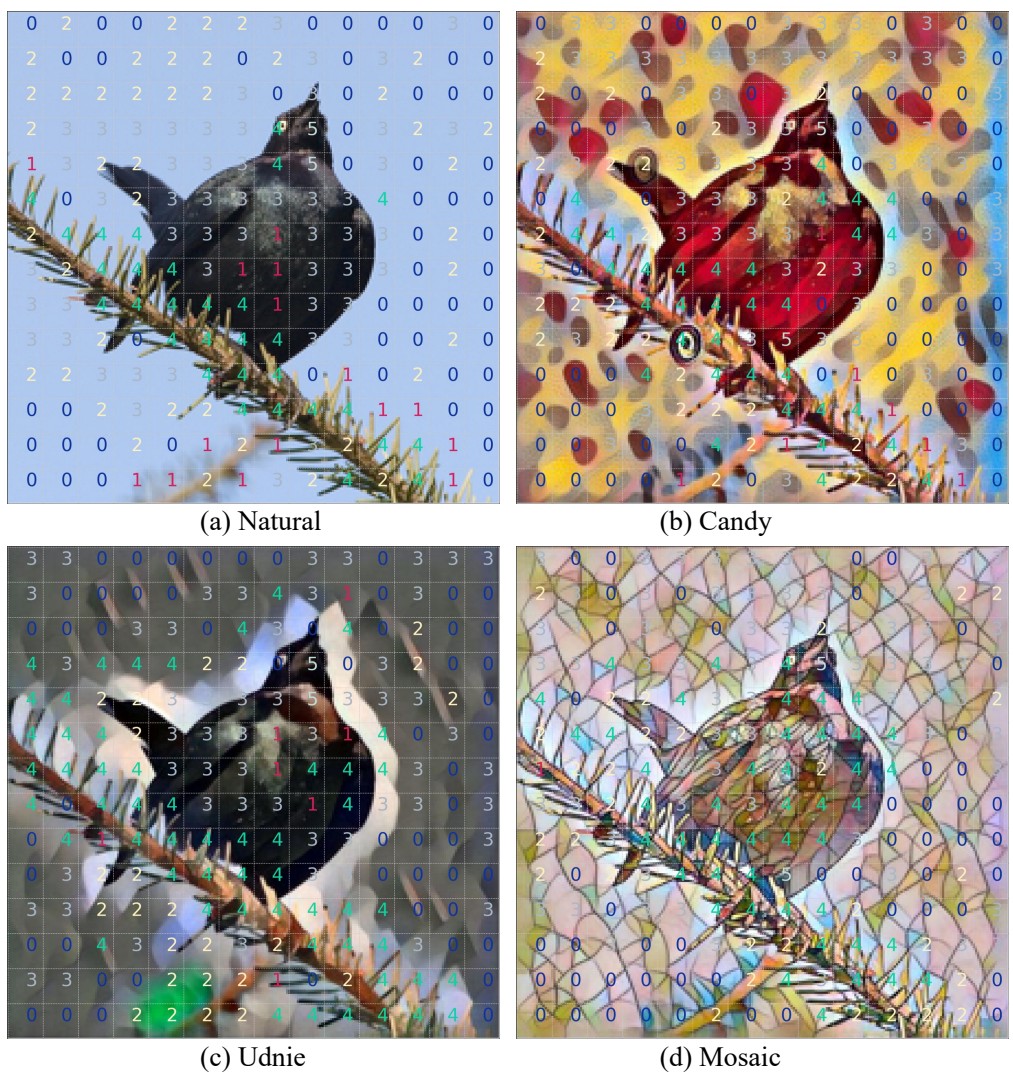

Figure 12: Expert selection visualization on block-10 of GMoE-S/16. Images are from different domains on CUB-DG.

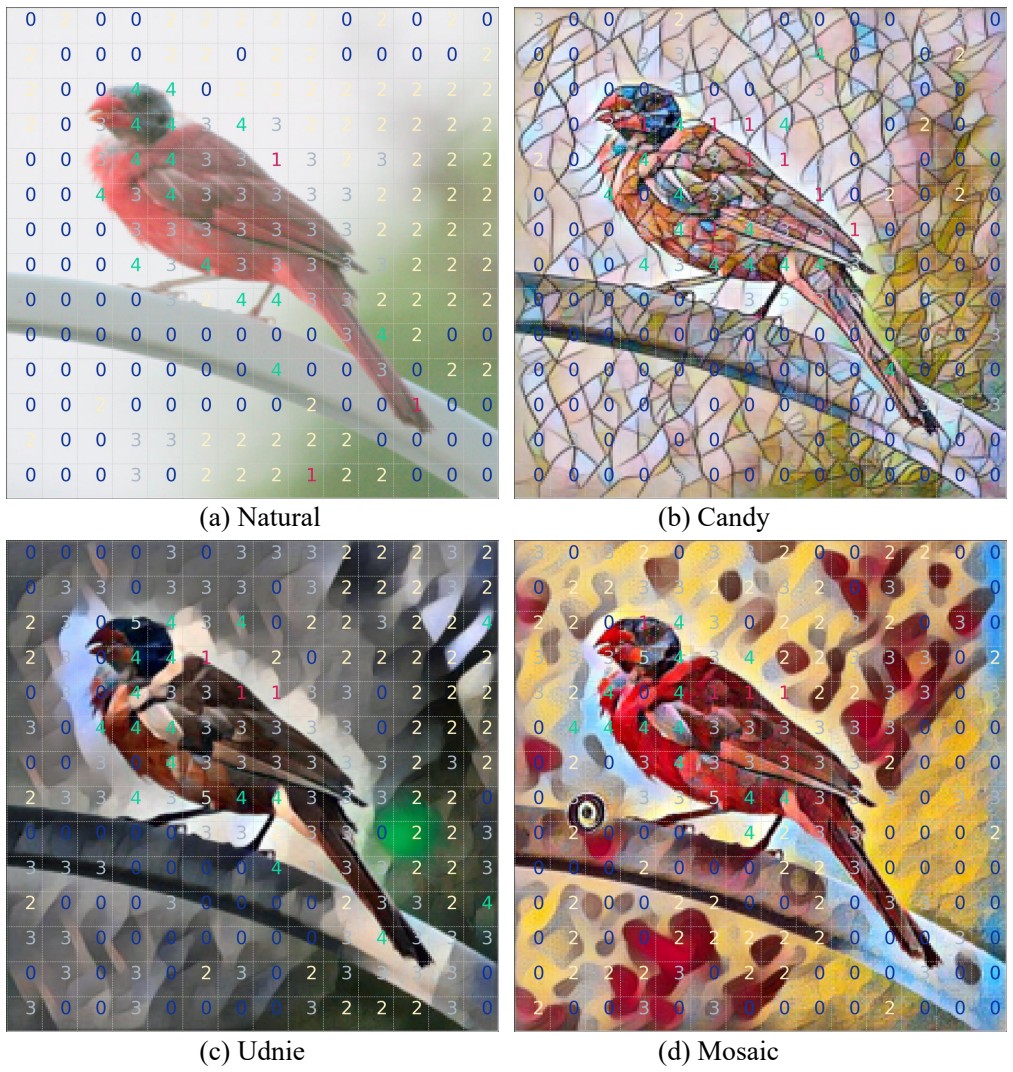

Figure 13: Expert selection visualization on block-10 of GMoE-S/16. Images are from different domains on CUB-DG.

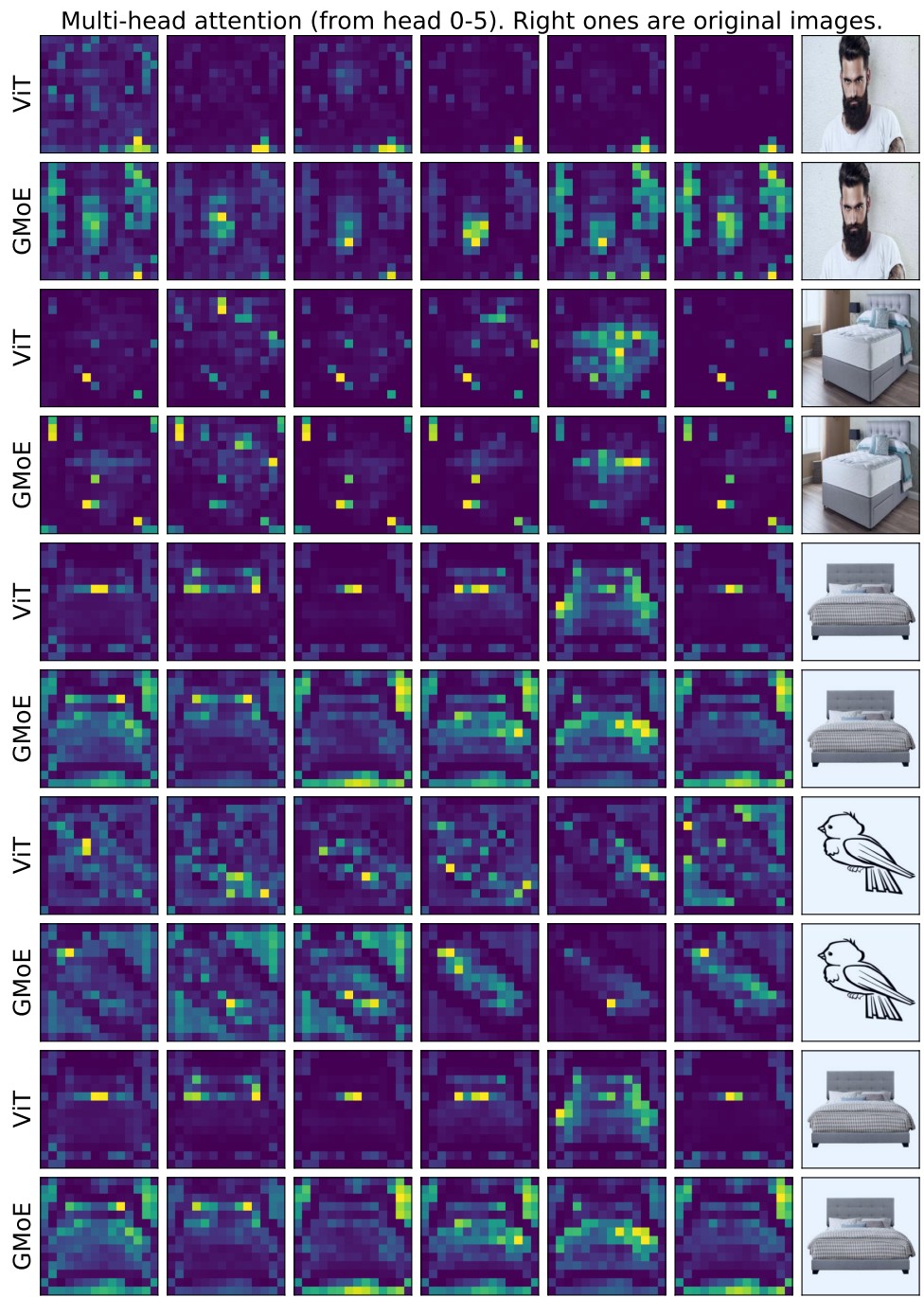

Figure 14: Multi-head attention visualization on the last block of ViT-S/16 and GMoE-S/16. Images are from *Real* domain in DomainNet.

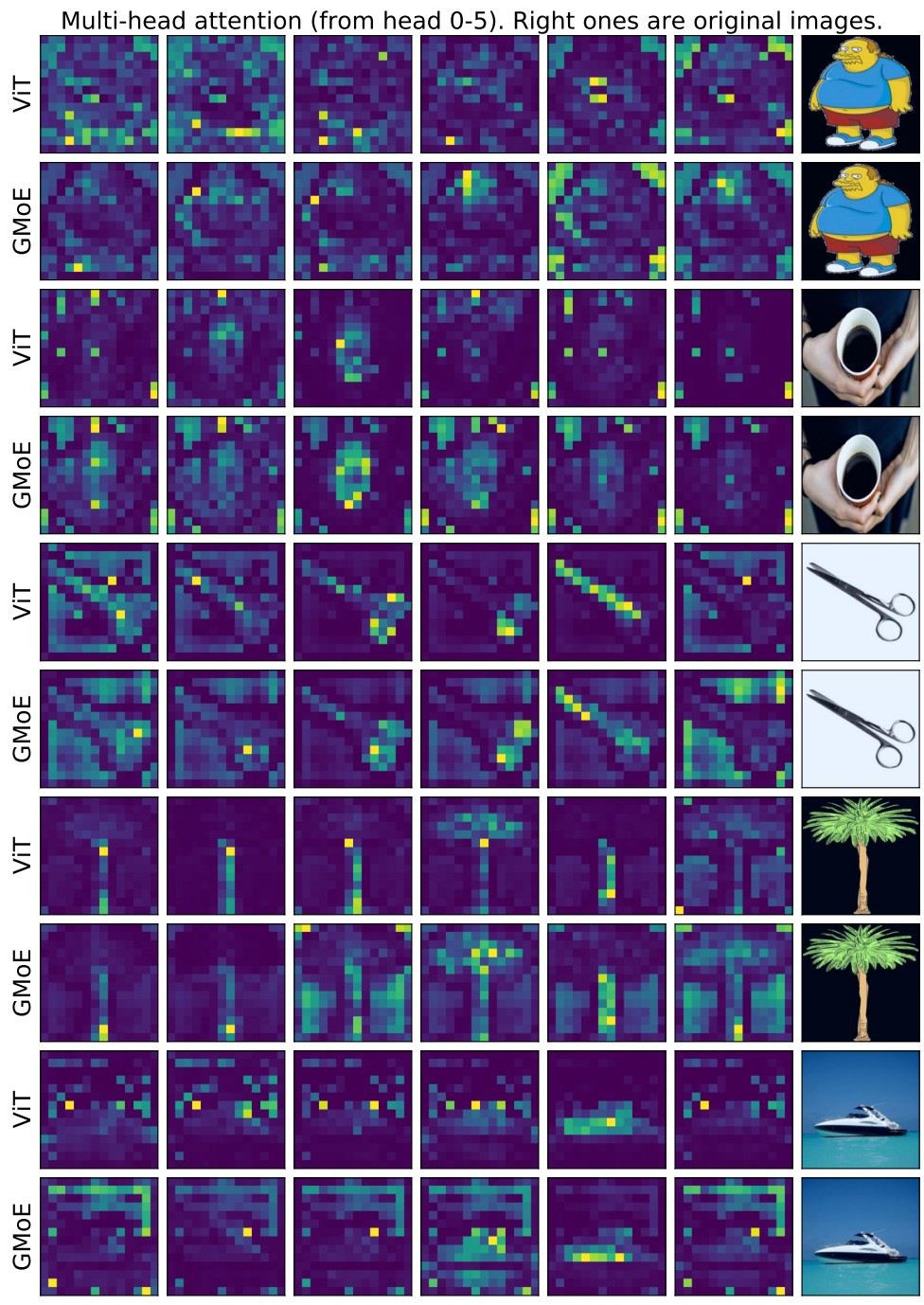

Figure 15: Multi-head attention visualization on the last block of ViT-S/16 and GMoE-S/16. Images are from *Real* domain in DomainNet.

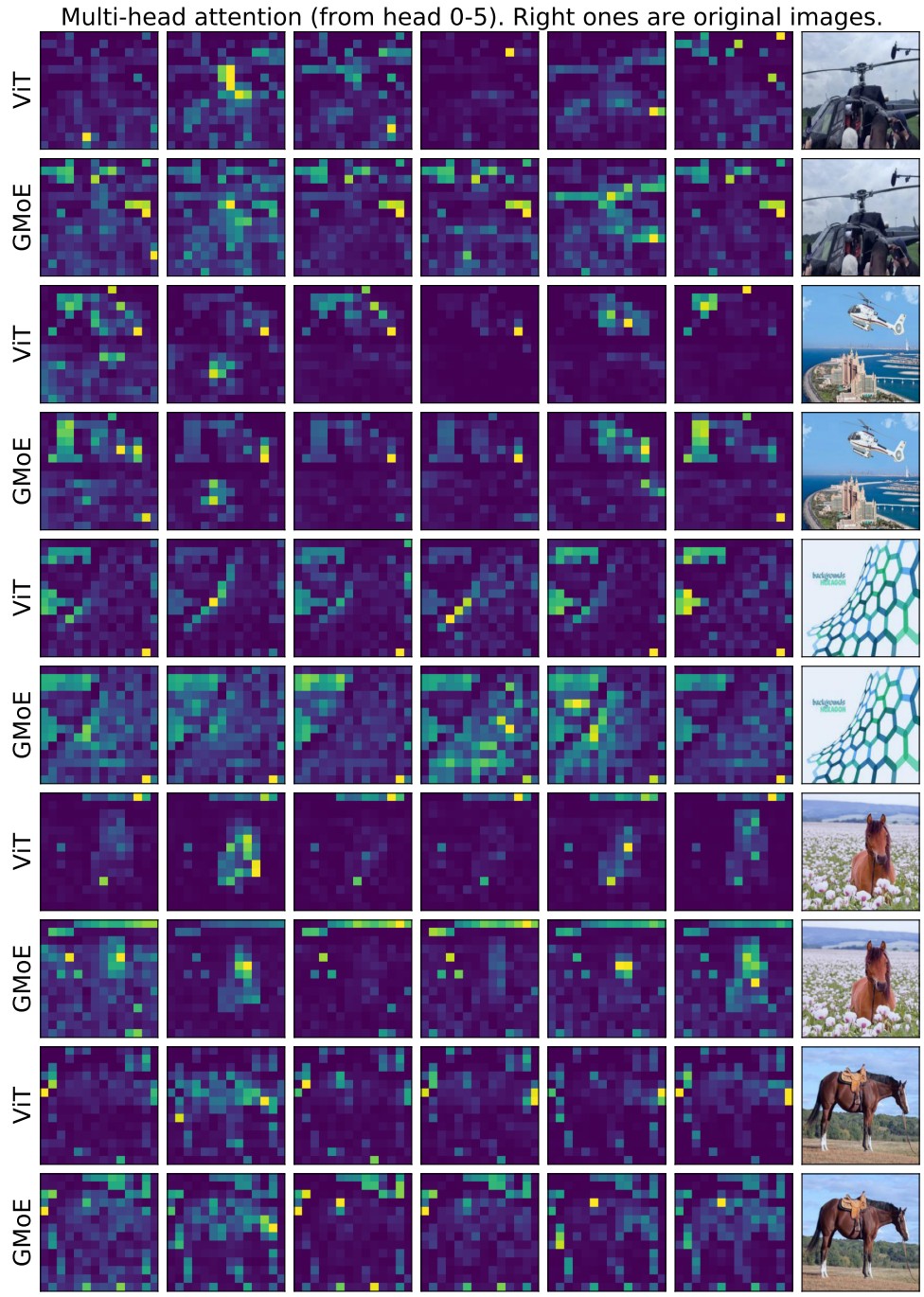

Figure 16: Multi-head attention visualization on the last block of ViT-S/16 and GMoE-S/16. Images are from *Real* domain in DomainNet.

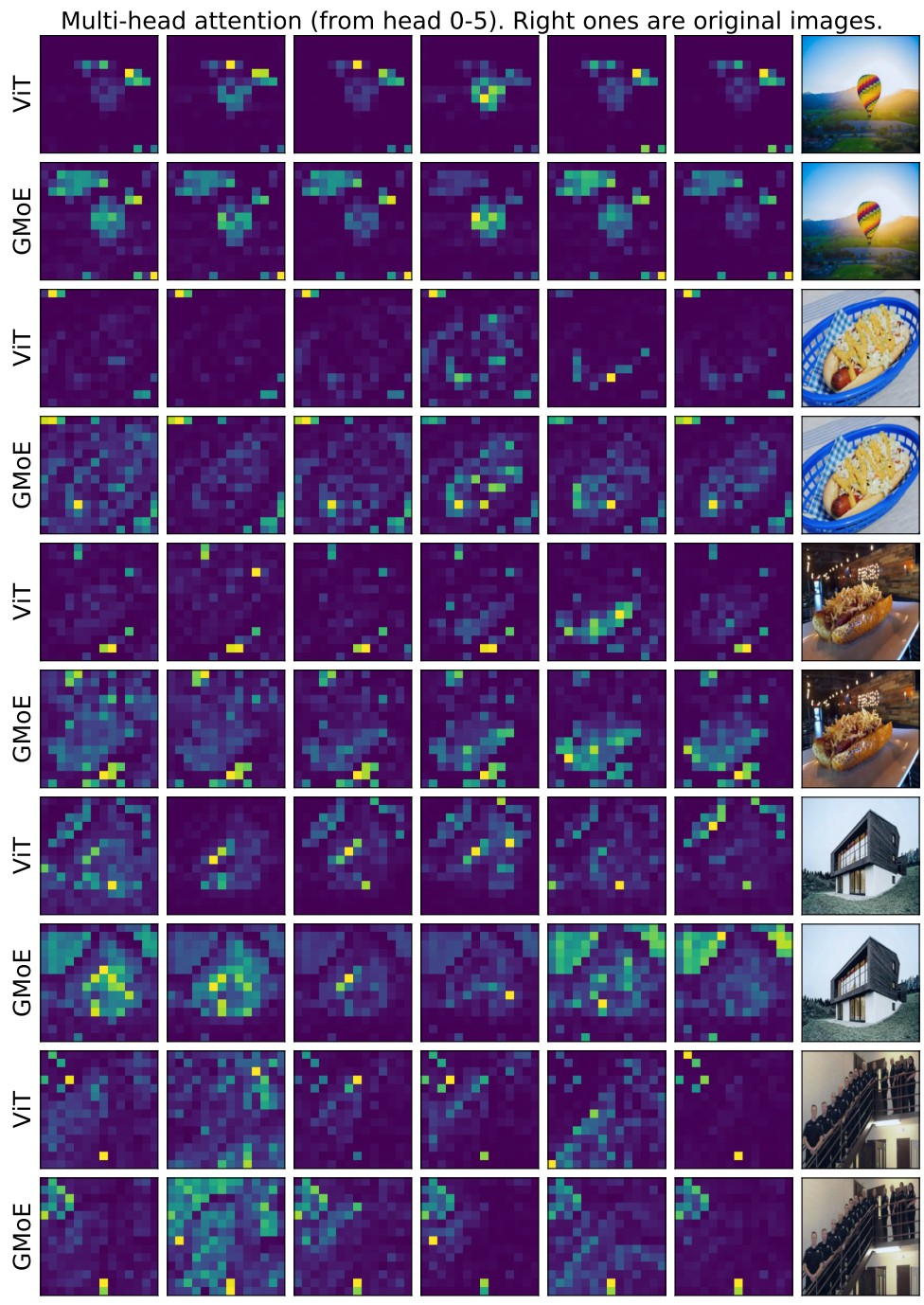

Figure 17: Multi-head attention visualization on the last block of ViT-S/16 and GMoE-S/16. Images are from *Real* domain in DomainNet.

