# OpenReview forum: "Sparse Mixture-of-Experts are Domain Generalizable Learners"
_ICLR.cc/2023/Conference — ICLR 2023 notable top 5%_

### Official Review · Reviewer_aHin · 2022-10-23

**Confidence:** 3
**Correctness:** 4
**Technical Novelty And Significance:** 3
**Empirical Novelty And Significance:** 3
**Recommendation:** 8

**Clarity, Quality, Novelty And Reproducibility:**

The paper is clearly written and hence easy to understand, although some of the definitions and theorems require careful reading. NB, I did not check the theorem proofs in detail.
Outside of the theoretical justification for the proposed architecture, none of the single architecture choices are novel per se - although in combination they attain sufficient novelty for the application to the domain generalization problem.
Code is provided for reproducibility, although I am unable to check this in practice.

**Strength And Weaknesses:**

As a clear strength, the paper provides detailed argumentation to motivate each of its design choices, going beyond a purely empirical study. Most prominently, it motivates the choice of ViT architecture by means of Theorem 1 and the choice of (G)MoE by means of Theorem 2. Both theorems provide a somewhat theoretical grounding of the architectural choices made. The empirical evaluation is extensive and convincing. A minor weakness is that further tweaks in Section 4.3 are made on purely empirical grounds and ablations are relegated to the appendix.

**Summary Of The Paper:**

The submission considers the problem of improving domain generalization via NN architecture design. The authors propose to adopt an MoE ViT design. The former (MoE) is motivated by its ability to serve conditional computation for easier processing of high-level visual attributed, while the latter (ViT) is motivated by multi-head self-attention properties as low-pass filter with shape bias compared to texture-biased CNNs. Routing scheme (cosine routing) and number of MoE layers are empirically adapted for best generalization performance. Experiments demonstrate superior performance over explicit domain generalization algorithms using more conventional CNN-based architectures.

**Summary Of The Review:**

The paper suggests well-grounded architecture choices to improve the SOTA in domain generalization using conditional computation via MoE.

---

> ### Author Response · Authors · 2022-11-15
> **Response to Reviewer aHin**
>
> We sincerely thank the reviewer for your insightful comments and recognition of this work, especially for acknowledging detailed argumentation to motivate each of its design choices. We have polished paper, included more intuitive explanations, and added more expressive visualizations.
>
> > A minor weakness is that further tweaks in Section 4.3 are made on purely empirical grounds and ablations are relegated to the appendix.
> >
>
> Thank you for pointing out this issue. According to the argument and theoretical analysis in Section 3 and Section 4.2, the network’s performance would be better if the MoE handles the visual attributes more efficiently. Section 4.3 designs the MoE components by following this guidance, though no theoretical analysis is provided. For example, the cosine router aligns with the codebook and matched filter to capture the visual attributes. It is important for future works to prove the operations (e.g. embedding normalization in cosine router) in Section 4.3 could improve the alignment. And we will move the ablations to the main text if space permits in our final version.

---

> > ### Comment · Reviewer_aHin · 2022-11-21
> > **Thank you**
> >
> > Thank you, sounds good to me. My recommendation stays the same.
> > Regards,
> > Reviewer aHin

---

### Official Review · Reviewer_FUc5 · 2022-10-24

**Confidence:** 4
**Clarity, Quality, Novelty And Reproducibility:** See above.
**Correctness:** 2
**Technical Novelty And Significance:** 3
**Empirical Novelty And Significance:** 3
**Recommendation:** 5

**Strength And Weaknesses:**

The paper focus on an interesting aspect and the paper are overall easy to understand. But some important experiments are missing.

1. Missing comparison between transformer with ERM and transformer with DG. The core motivation "empirical
risk minimization (ERM) outperform CNN-based models employing state-of-the-art (SOTA) DG algorithms" can just because the transformer is a stronger backbone.

2. Why is GMoE superior to a normal MoE.
The major difference is that it forces a normalized operation on the input of the gating function and uses a learned embedding dictionary to decide the gating. The ablation study of why this is superior is missing from the paper.

3. The paper states that GMoE relies on visual attributes to have better performance. This is not convinceable from figure 3(b). It seems that e0 and e2 are specialized for background and others (e3,e4, e5) are all activated by almost all the visual attributes. Figure 3(c) is also not convincing to me. It seems no strong relation between experts and attributes. Any network would have similar visualization that some part of it prefers some region in the image.

Overall, the experiment parts seem to be a weakness: not enough experiments, ablations, and convincing visualization.

**Summary Of The Paper:**

The paper start from an interesting empirical finding that a transformer trained with ERM outperforms CNN trained with domain generalization (DG) algorithms on DG task. They show some experiments and math to prove that. They propose a new GMoE and show its superiority on DG.

**Summary Of The Review:**

See above.

---

> ### Author Response · Authors · 2022-11-15
> **Response to Reviewer FUc5 (part 2)**
>
> > Why is GMoE superior to a normal MoE. The major difference is that it forces a normalized operation on the input of the gating function and uses a learned embedding dictionary to decide the gating. The ablation study of why this is superior is missing from the paper.
> >
>
> Thank you for pointing out this important issue. Due to space limitation, the ablation study is not emphasized in the main text. The comparison between linear gating and cosine gating is given in the following table and more ablations on MoE architecture design are given in Appendix D.5. The normalized operation in cosine router outperforms linear gating by $0.8\%$ on average, which is a non-trivial improvement on DomainBed benchmark. From the table, we see that cosine router is useful in MoE for DG.
>
> |  | Configuration | Router | PACS | VLCS | OfficeHome | TerraInc | DomainNet | Avg |
> | :---: | :---: | :---: | :---: | :---: | :---: | :---: | :---: | :---: |
> | ViT-S/16 ERM | - | - | 86.2 | 79.7 | 72.2 | 42.0 | 47.3 | 65.5 |
> | GMoE-S/16 ERM | Last 2 | Linear | 87.8 | 80.0 | 72.7 | 46.7 | 48.3 | 67.1 |
> | GMoE-S/16 ERM | Last 2 | Cosine | 88.1 | 80.2 | 74.2 | 48.5 | 48.7 | 67.9 |
> |  |  |  |  |  |  |  |  |  |
>
> > The paper states that GMoE relies on visual attributes to have better performance. This is not convinceable from figure 3(b). It seems that e0 and e2 are specialized for background and others (e3, e4, e5) are all activated by almost all the visual attributes. Figure 3(c) is also not convincing to me. It seems no strong relation between experts and attributes. Any network would have similar visualization that some part of it prefers some region in the image.
> >
>
> Thank you for pointing this important issue out. We realized that Fig. 3(c) is not clearly illustrated, and we have replaced it in the revised manuscript (one may also refer to the anonymous links [https://imgur.com/a/wdv3ciC](https://imgur.com/a/wdv3ciC) and [https://imgur.com/a/Uu2X89d](https://imgur.com/a/Uu2X89d) for previewing the figures).
>
> Our major contribution is to theoretically investigate the impact of backbone architecture in DG and repurpose MoE for DG, where the visualization is a by-product. As the labeling in CUB dataset is not comprehensive and there is no supervision signal for visual attributes, the correlations between experts and attributes cannot be perfect.
>
> For Fig. 3(b), although e3, e4, e5 are activated by many visual attributes, a specific expert has a much stronger preference for some attributes over others. Specifically, e3 focuses on the wings, tail, back, and beak; e4 focuses on the legs, breast, and belly. In Fig. 3(c), the areas around the wings activate e3, and the areas around the legs activate e4.
>
> In addition, the birds in CUB locate in different regions of the image and have different orientations. If the experts are correlated to the regions, the correlations in Fig. 3(b) should be uniformly distributed. However, the quantitative results in Fig. 3(b) show that the experts have strong correlations to certain attributes, which indicates that experts are not correlated to the regiond.
>
> In Appendix E of revised manuscript, we have also included two parts of visualization results with more expressive images.
>
> (1) We provide expert selection results in natural domain images across different bird classes, i.e., Fig. 8 and Fig. 9. We use different colored lines that correspond to different visual attributes, and we connect these lines across images in a row. As seen, this part of the results may indicate that experts always tend to focus on the same visual attributes while seeing different bird images from different angles. This shows that the correlations between experts and attributes are potentially invariant across classes.
>
> (2) In order to more vividly display the expert selection across domains, we use different colors to highlight different selected experts, i.e., Fig. 10 - 13. We see that, in most cases, the expert selection is consistent across domains. The results reveal that experts are potentially invariant in dealing with visual attributes across domains.

---

> ### Author Response · Authors · 2022-11-15
> **Response to Reviewer FUc5 (part 1)**
>
> We sincerely thank the reviewer for your insightful and constructive feedbacks. **We have added the experiments, ablation studies, and clear visualizations in the revised manuscript.**
>
> Specifically, in Appendix D.4 of our revised manuscript, we compare GMoE, ViT and ResNet architecture with recent DG algorithms (e.g. Swad, Fish, Fishr). In Appendix D.6, we show ablation study on MoE architecture design, and Appendix E contains more clear visualization results.
>
> ---
>
> > *Missing comparison between transformer with ERM and transformer with DG.*
> >
>
> We thank the reviewer for helping us strengthen the experiments. In the original manuscript, we only investigated the performance of the transformer with ERM in Table 1 while missing the experiments of the transformer with recent DG algorithms.
>
> According to the comments, we have added more experiments for the transformer with recent DG algorithms. The results are shown in the following table, where GMoE consistently outperforms ViT, as well as ResNet.
>
> |  | PACS | VLCS | OfficeHome | TerraInc | DomainNet | Avg |
> | :---: | :---: | :---: | :---: | :---: | :---: | :---: |
> | ViT-16/S SWAD | 88.3 | 80.4 | 72.7 | 46.9 | 48.8 | 67.4 |
> | GMoE/S SWAD | 88.6 | 80.8 | 74.5 | 49.1 | 49.6 | 68.5 |
>
> ---
>
> |  | PACS | VLCS | OfficeHome | TerraInc | DomainNet | Avg |
> | :---: | :---: | :---: | :---: | :---: | :---: | :---: |
> | ResNet50 Fish | 85.5 | 77.8  | 68.6 | 45.1 | 43.1 | 64.0 |
> | ViT-16/S Fish | 87.3 | 78.0 | 72.2 | 42.7 | 47.3 | 65.5 |
> | GMoE/S Fish | 87.9 | 79.7 | 73.0 | 46.9 | 48.8 | 67.3 |
> | ResNet50 Fishr | 85.5 | 77.8 | 68.6 | 47.4 | 41.7 | 64.2 |
> | ViT-16/S Fishr | 85.6 | 79.1 | 71.8 | 44.1 | 47.6 | 65.6 |
> | GMoE-16/S Fishr | 88.3 | 78.8 | 72.7 | 45.6 | 48.3 | 66.7 |
>
> ---
>
> > *The core motivation "empirical risk minimization (ERM) outperform CNN-based models employing state-of-the-art (SOTA) DG algorithms" can just because the transformer is a stronger backbone.*
> >
>
> Thanks for helping us clarify the motivations by this interesting question. We have added new experiments to understand whether the superior OOD performance comes from the in-domain performance gain. Specifically, we adopt ResNet-50 V2, which adopts strong augmentation and achieves better performance than ViT-S/16 and GMoE-S/16 on ImageNet-1K.  The training recipes are shown in the following table and the details are given in Appendix D.7.
>
> | Recipe | LR Opt | Trival Augumentation | Random Erasing | Label Smoothing | FixRes Mitigations | WR tuning | IN1K |
> | :---: | :---: | :---: | :---: | :---: | :---: | :---: | :---: |
> | ResNet50 V2 | ✔ | ✔ | ✔ | ✔ | ✔ | ✔ | 80.8 |
> | ViT-S/16 | ✔ | ✗ | ✗ | ✗ | ✗ | ✗ | 79.9 |
> | GMoE-S/16 | ✔ | ✗ | ✗ | ✗ | ✗ | ✗ | 80.1 |
>
> ---
>
> We test ResNet50 V2 on three datasets in Fig. 1 and the results are shown in the following table. Though Resnet-50 V2 achieves better performance on ImageNet, it is still outperformed by ViT/S on VLCS, OfficeHome, and DomainNet, which is consistent with the results in Fig. 1. The results show that a stronger IID performance does not imply better OOD performance and the backbone architecture plays a pivotal role in DG, verifying the statements of Theorem 1.
>
> |  | VLCS | OfficeHome | DomainNet |
> | :---: | :---: | :---: | :---: |
> | ResNet50 V2 ERM | 78.2 | 68.7 | 45.3 |
> | ResNet50 V2 Fishr | 77.9 | 70.4 | 47.0 |
> | ViT-S/16 ERM | 79.7 | 72.2 | 47.1 |
> | GMoE-S/16 ERM | 80.2 | 74.2 | 48.7 |

---

> ### Author Response · Authors · 2022-11-24
> **Sincerely Looking Forward to Your Reply**
>
> Dear Reviewer,
>
> Your suggestions and comments have greatly helped polish our paper, from the ablation studies (Q1, Q2) and visualization (Q3). We sincerely look forward to your reply to our response, and we are open to any discussion to improve our paper.
>
> Best wishes,
>
> The authors.

---

### Official Review · Reviewer_jTSg · 2022-10-25

**Confidence:** 3
**Correctness:** 4
**Technical Novelty And Significance:** 3
**Empirical Novelty And Significance:** 3
**Recommendation:** 8

**Clarity, Quality, Novelty And Reproducibility:**

The paper is clear and has good quality. The proposed method is not original, it repurposes similar methods already used in the community but it applies them in a different setting (of domain generalization) and motives their good performance by the formalism of alignment.

**Strength And Weaknesses:**

**Strengths**:

The paper tackles an important problem of domain generalization and proposes a good direction to approach it, by designing models that are well aligned with the desired task.

The formalization of alignment between models and the task is sound and useful for guiding the design of future architectures.

The proposed model seems to be beneficial for DG tasks. It is good to see that GMoE obtains good results with standard ERM training, but also benefits from DG algorithms.

It is interesting to see that the experts seem to specialize, as seen by histograms and examples in Section 5.

**Weaknesses**

The paper did not explain in much detail why conditional statements are good for Domain Generalisation. As this point is crucial in linking the proposed method to DG, it should be better explained. At the high-level Theorem 2 says that the proposed model is good for cases where the true function is composed of similar mechanisms (selection + application of submodule) as the GMoE model (gating + separate FFN). But still, it should be explained in more detail why such functions are preferable for DG. Some intuitions are given in Section 4.2, but expanding them would be good.

The paper could discuss connections to other works that mixture-of-experts for domain adaptation [A], or use experts in similar ViT models [B].


[A] Guo et al. “Multi-Source Domain Adaptation with Mixture of Experts” EMNLP 2018
[B] Rahaman et al. "Dynamic inference with neural interpreters." NeurIPS 2021.


**Summary Of The Paper:**

This paper tackles the problem of domain generalization by proposing to focus on model architectures that are well aligned to the task of interest. The paper uses the definition of algorithmic alignment of Xu et al. 2020 which defines how easy it is to learn a function that can be decomposed into N subfunctions if your neural network can also decompose into similar subcomponents. First, the paper formally shows that models that align well with the causal correlations in the data will generalize well, while models that align well with the spurious correlations will not generalize. The authors propose a model named Generalizable Mixture-of-Experts (GMoE), similar to other mixture-of-experts Transformer models used primarily in NLP. The authors argue that domain generalization benefits from capturing and effectively combining diverse attributes and for this purpose, some kind of conditional statements are required. It is formally shown that such statements align well with the proposed model, thus the GMoE model should generalize well out-of-distribution. Experimentally, the GMoE is shown to obtain good results compared to recent methods.


**Summary Of The Review:**

Overall the paper is well-written and has a good formalism that gives theoretical explanations of the good performance of the proposed method. The paper will benefit the community by showing that good architectures that align with the task are to be desired and show a good example of such architecture.

---

> ### Author Response · Authors · 2022-11-15
> **Response to Reviewer jTSg**
>
> We sincerely thank the reviewer for your insightful comments and recognition of this work, especially for acknowledging our theoretical contributions. We have polished the paper and made the clarifications in the revised version.
>
> > The paper did not explain in much detail why conditional statements are good for Domain Generalisation. As this point is crucial in linking the proposed method to DG, it should be better explained. At the high-level Theorem 2 says that the proposed model is good for cases where the true function is composed of similar mechanisms (selection + application of submodule) as the GMoE model (gating + separate FFN). But still, it should be explained in more detail why such functions are preferable for DG. Some intuitions are given in Section 4.2, but expanding them would be good.
> >
>
> Conditional statements can be understood as if/else statements in programming. Suppose we train the network to recognize the elephants on DomainNet, as illustrated in the first row of Fig. 1(b). For the elephants in different domains, shape and texture vary significantly while the visual attributes (large ears, curved teeth, long nose) are invariant across all the domains. Equipped with conditional statements, the recognition of the elephants can be expressed as “if an animal has large ears, two curved outer teeth, and a long nose, then it is an elephant”. Then the subtasks are to recognize these visual attributes, which also requires conditional statements. For example, the operation for “curved outer teeth” is that “if the patch belongs to the teeth, then we apply a shape filter to it”. Therefore, conditional statements play crucial roles in recognizing the same object across different domains. Although conditional statements can be learned via FFN (as MLPs are universal approximations), Theorem 2 suggests that MoE learns them easier and leads to a better generalization performance. We have revised the paper accordingly to reflect these intuitions.
>
> > The paper could discuss connections to other works that mixture-of-experts for domain adaptation [A], or use experts in similar ViT models [B]. [A] Guo et al. “Multi-Source Domain Adaptation with Mixture of Experts” EMNLP 2018 [B] Rahaman et al. "Dynamic inference with neural interpreters." NeurIPS 2021.
> >
>
> Thanks for bringing our attention to these important references and we have added discussions in related works. [Ref. A] considers the muti-source domain adaptation in NLP, where one classifier is trained on each domain and a mixture-of-experts is adopted to ensemble the classifiers. [Ref. B] investigates the systematic generalization problem and propose to dynamically compose experts instead of selecting experts in MoEs. In this paper, we consider the domain generalization (DG) problem and prove that a backbone more aligned to invariant correlations is more robust to distribution shift. Based on the theory, MoE is adopted to replace FFN for enhancing alignment and thus improving DG performance.
>
> ---
> [Ref. A] Guo et al. “Multi-Source Domain Adaptation with Mixture of Experts” EMNLP 2018
>
> [Ref. B] Rahaman et al. "Dynamic inference with neural interpreters." NeurIPS 2021.

---

### Official Review · Reviewer_nMjY · 2022-10-27

**Confidence:** 5
**Correctness:** 3
**Technical Novelty And Significance:** 3
**Empirical Novelty And Significance:** 3
**Recommendation:** 6

**Clarity, Quality, Novelty And Reproducibility:**

The paper is ok to read. However, the theorems can be better connected with the text and experiments. Novelty is primarily on the GMoE experiments and efforts to connect them with some intuition. The works appears reproducible.

**Strength And Weaknesses:**

Strengths

- I liked the idea of applying mixture of experts for solving the problem of DG. I resonate with the authors, that it is a bright direction for explorations in DG
- The experiments are thorough and the results appear promising and t

Weaknesses

My major concern with the paper is that some important recent papers or results/discussions from cited papers are missing.

(a) For instance, why did the authors ignore the MIRO+SWAD numbers in Cha et al. 2022. It clearly and comprehensively outperforms the proposed method on all the datasets.

(b) Author missed in important recent work by Sivaprasad et al. [A], which makes a similar observation that backbone plays a more crucial role in DG compared to tailored algorithms. They show that ERM with Inception Resnet backbone can achieve 89.11% accuracy on PACS (outperforming the proposed approach on the particular dataset). That clearly dilutes the first contribution of the paper. I suspect more similarities with that work as well in terms of distribution shifts etc. A discussion is warranted.

(c) Finally, were the experiments in Table 1 averaged over multiple runs?


[A] Sivaprasad et al. Reappraising Domain Generalization in Neural Networks, arXiv 2021




**Summary Of The Paper:**

The paper proposes a GMoE model for solving classification under domain shifts, specifically in the DG setting. The paper presents results on 8 benchmarks and shows promising performance.

**Summary Of The Review:**

My primary concerns are on missing results and papers and would request the authors to respond to it in the rebuttal period.

---

> ### Author Response · Authors · 2022-11-15
> **Response to Reviewer nMjY**
>
> We sincerely thank the reviewer for your insightful comments and recognition of this work, especially for acknowledging that employing mixture-of-experts is a bright direction for DG. We have polished the paper, added the experiments, and clarified the below points in the revised manuscript.
>
> > *(a) My major concern with the paper is that some important recent papers or results/discussions from cited papers are missing. For instance, why did the authors ignore the MIRO+SWAD numbers in Cha et al. 2022. It clearly and comprehensively outperforms the proposed method on all the datasets.*
> >
>
> We thank the reviewer for pointing this important issue out. Due to space limitations, we did not explain the choice of baselines in detail, which may cause some confusion. To make a fair comparison, the competing methods should have a similar number of parameters/execution time and be trained on the same pretraining dataset. In Table 1 of our paper, methods are pretrained on ImageNet-1K with **1.28 million training samples** (following [Ref. B]). The backbone adopted in “Using RegNetY-16GF backbone with SWAG pre-training” (Table 1 of [Ref. A]) is pretrained on IG-3.6B with **10 billion training samples**. Therefore, we did not include the accuracy achieved by “Using RegNetY-16GF backbone with SWAG pre-training”.
>
> Additionally, the baselines in Table 1 follow DomainBed [Ref. B], which focuses on single DG algorithms, as any two orthogonal DG algorithms can be combined together to improve performance. We have also added the experiments to compare GMOE+SWAD with MIRO+SWAD pretrained on ImageNet-1K. The results are shown in the following table, where the proposed method outperforms MIRO on 4/5 datasets.
>
> |  | PACS | VLCS | OfficeHome | TerraInc | DomainNet |
> | :---: | :---: | :---: | :---: | :---: | :---: |
> | MIRO | 85.4 | 79.0 | 70.5 | 50.4 | 44.3 |
> | GMoE (Ours) | 88.1 | 80.2 | 74.2 | 48.5 | 48.7 |
> | MIRO+SWAD | 88.4 | 79.6 | 72.4 | 52.9 | 47.0 |
> | GMoE+SWAD (Ours) | 88.6 | 80.8 | 74.5 | 49.1 | 49.6 |
>
> > *(b) Author missed in important recent work by Sivaprasad et al. [A], which makes a similar observation that backbone plays a more crucial role in DG compared to tailored algorithms. That clearly dilutes the first contribution of the paper. I suspect more similarities with that work as well in terms of distribution shifts etc. A discussion is warranted.*
> >
>
> Thanks for bringing our attention to this important reference. **We have added it as a reference in the revised manuscript and discussed it in related works.** [Ref. C] investigates the effect of the optimizer, augmentation, and backbone on DG performance. Specifically, in Section 4.2 of [Ref. C], experiments were conducted to test $6$ backbones on PACS dataset, among which Inc-Resnet performs the best.
>
> Nevertheless, it did not explain when a specific model will succeed in DG and how to design the backbone architecture to improve DG performance. As far as we know, our paper is an initial attempt to theoretically investigate these questions, which provides a novel perspective of DG. In this paper, we prove that a backbone is more robust to distribution shift if it aligns with the invariant correlation in the dataset and a better alignment leads to better performance in DG. A novel architecture is designed based on our theory. We believe these results will encourage further research in this direction.
>
> > *They show that ERM with Inception Resnet backbone can achieve 89.11% accuracy on PACS (outperforming the proposed approach on the particular dataset).*
> >
>
> To make a fair comparison, the competing methods should have similar computational cost. Inc-Resnet is of 6.3 GMACs, which is 31.3% higher than GMoE (4.8 GMACs). Nevertheless, GMoE trained with ERM significantly outperforms ERM-Inc-Resnet on the other three datasets, showing the generalizability of the proposed architecture.
>
> |  | Computation Load (GMACs) ↓ | PACS ↑ | VLCS ↑ | OfficeHome ↑ | DomainNet ↑ | Avg ↑ |
> | :---: | :---: | :---: | :---: | :---: | :---: | :---: |
> | ERM-Inc-Resnet | 6.3 | 89.1 | 78.8 | 72.0 | 43.2 | 70.8 |
> | ERM-GMoE | 4.8 | 88.1 | 80.2 | 74.2 | 48.7 | 72.8 |
>
> > *(c) Finally, were the experiments in Table 1 averaged over multiple runs?*
> >
>
> We appreciate your valuable comments and we have clarified it in Section 5. In the original manuscript, the detailed experiment setting follows [Ref. B] and left to Appendix D1 due to space limitation. The experiments in Table 1 are averaged over 3 runs as in [Ref. B].
>
> ---
> [Ref. A] Junbum Cha, et al. “Domain Generalization by Mutual-Information Regularization with Pre-trained Models.” ECCV 2022.
>
> [Ref. B] Ishaan Gulrajani and David Lopez-Paz. In search of lost domain generalization. ICLR 2021.
>
> [Ref. C] Sarath Sivaprasad, et al. Reappraising Domain Generalization in Neural Networks, arXiv 2021

---

> > ### Comment · Reviewer_nMjY · 2022-11-21
> > **Rating post author rebuttal**
> >
> > The author response is satisfactory and addresses the concerns raised in my review. I am increasing my rating to account for the same.

---

### Author Response · Authors · 2022-11-15
**General Response**

### General Response

We sincerely thank all the reviewers for your constructive feedbacks and recognitions of this work, especially for acknowledging that **the novelty of formal investigation of backbone architecture in DG** (Reviewer jTSg, aHin), **the novelty for repurposing MoE for DG** (Reviewer jTSg, FUc5, aHin), **the performance is superior** (all reviewers), and **the experiments are thorough** (Reviewer nMjY, aHin).

We have polished the paper, tried our best to complete all experiments asked by reviewers, and made the clarifications in the revised version.

We would like to re-emphasize the novelty and technical contributions of this work:

1. To the best of our knowledge, **this work is an initial attempt to formally investigate the impact of backbone architecture in DG under a theoretical framework**, which is orthogonal to the previous works. **The theory established in this paper explains the superior performance of vision transformer-based models and guides novel architecture design of our proposed GMoE for performance enhancement.** We sincerely hope that our contributions could be appreciated.
2. Based on theoretical analysis, we propose a generalizable mixture-of-experts (GMoE) for DG, built upon vision transformers. **1)** We repurpose MoE from scaling large networks to efficiently process invariant visual attributes in DG; **2)** The modifications are developed to adapt the original MoE to DG setting, i.e., cosine router and last-two configuration. **3)** We prove that GMoE enjoys a better alignment than vision transformers. The effectiveness of MoE modules and modifications is verified by extensive experiments on DomainBed.
3. As an early attempt to explore the backbone design in DG, this work could pave the way for further studies in this community. Specifically, our approach gives an idea to design backbones for DG based on architecture-task alignment. We hope this paper could provide insights into relevant domains like robustness and few-shot learning.

We have revised our manuscript to include the following changes according to all the reviewers’ insightful comments. Note that all the polishments on the main submission and supplemental document are highlighted with rose red text color for better visualization.

1. We have included an intuitive example to demonstrate the use of conditional sentences in DG in Section 4.2.
2. We have replaced Fig. 3(c) with a more expressive figure that highlights visual attributes and added more illustrative examples in Appendix E (Fig. 8-13).
3. We have added a detailed comparison of GMoE with DG algorithms and transformer with DG algorithms in Appendix D.4.
4. We have added a discussion for papers that investigate the robustness on distribution shifts in related works in Appendix A.6.

Due to the constraints of time and computational resources in the rebuttal period, all our experiments in the rebuttal stage were only carried out in one random seed (instead of three in our main paper). We will update these results with three runs in the final version.

Please don't hesitate to let us know of any additional comments on the manuscript or the changes.

---

### Author Response · Authors · 2022-11-21
**Looking Forward to Further Discussions**

Dear reviewers,

Thank you again for your valuable time and insightful comments! We sincerely look forward to your reply to our response to let us know if we have resolved your concerns, and we are open to any discussion to improve our paper.

Best regards!

The authors

---

### Decision · Program_Chairs · 2023-01-20

**Decision:**

Accept: notable-top-5%

**Justification For Why Not Higher Score:**

N/A

**Justification For Why Not Lower Score:**

N/A

**Metareview: Summary, Strengths And Weaknesses:**

This is a solid paper that investigates the impact of the backbone on domain generalization settings and proposes the Generalizable Mixture-of-Experts, a model build on vision transformed that addresses classification under domain shifts.
Results on 8 benchmarks show promising performance.
I personally find the idea very interesting and the paper well constructed, from analysis to method. I also believe it has a quite widespread applicability, potentially beyond vision tasks.

**Note From Pc:**

if the above contains the word "oral" or "spotlight" please see: "oral" presentation means -> notable-top-5% and "spotlight" means -> notable-top-25%. As stated in our emails, we are disassociating presentation type from AC recommendations